# Divide-and-Conquer Time Series Forecasting with Auto-Frequency-Correlation via Cross-Channel Attention

## Abstract

To model various short-term temporal variations, we propose an effective design of Transformer-based, termed FreCoformer. FreCoformer is designed on top of the frequency domain and comprises three key designs: frequency patching operation and two independent observations of these patches. The patching process refines the frequency information, enhancing the locality. The subsequent observations extract the consistent representation within different channels by attention computation and summarize the relevant sub-frequencies to identify eventful frequency correlations for short-term variations. To improve the data fit for different time series scenarios, we propose a divide-and-conquer framework and introduce a simple linear projection-based module, incorporated into FreCoformer. These modules learn both long-term and short-term temporal variations of time series by observing their changes in the time and frequency domains. Extensive experiments show the effectiveness of our proposal can outperform other baselines in different real-world time series datasets. We further introduce a lightweight variant of FreCoformer with attention matrix approximation, which achieves comparable performance but with much fewer parameters and computation costs. The code is available: `https://anonymous.4open.science/r/FreCoformer-6F22`

## 1 Introduction

Time series forecasting is an essential task in various applications and has recently witnessed great advancements powered by deep learning methods, especially Transformer (Zhou et al., 2021; Woo et al., 2022b; Nie et al., 2023; Wen et al., 2023). Such methods aim to discern consistent feature representations in historical observations and forecasting time series. Successful approaches usually involve learning representation in long-term temporal variations, e.g., trend and seasonality (Wen et al., 2020). These variations are typically extracted through time series decomposition (Woo et al., 2022a). Subsequently, they leverage the attention mechanism in Transformer to automatically learn temporal dependencies of these variations to yield consistent representations (Wen et al., 2023).

Nevertheless, these approaches inevitably lead to information loss of short-term temporal variations in some complex scenarios (Liu et al., 2022c; Wu et al., 2023). Figure 1(a) illustrates an electricity case where modeling long-term variations mainly captures low-frequency features, neglecting many consistent mid-to-high frequency components. Such components manifest as short-term variations, such as fluctuations and periodicities over short durations, and are good guidances for several practical analyses (Crespo Cuaresma et al., 2004; Thompson & Wilson, 2016; Hammond et al., 2023).

To end this, previous studies have leveraged frequency decomposition and spectrum information to assist Transformer in modeling temporal dependencies (Wu et al., 2021; Woo et al., 2022b). However, low-frequency components generally carry most of the energy in the spectrum and are dominant in real-world time series (Zhu & Shasha, 2002; Corripio et al., 2006). Influenced by such redundant low-frequency and noise, these approaches tend to prioritize long-term temporal variations (Figure 1(b)). Moreover, researchers directly deploy Transformers to the frequency domain to identify more eventfully relevant high-frequency components (Zhou et al., 2022). Despite enhancements in frequency attention, this approach relies on heuristic and empirical strategies, i.e., *random*

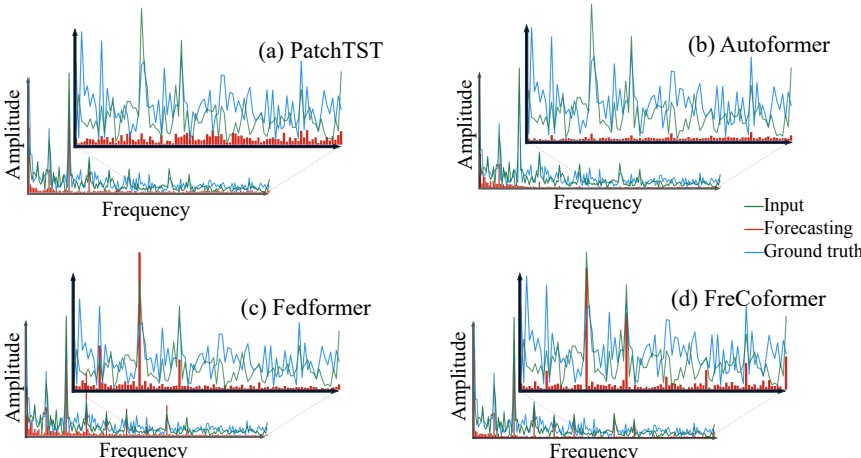

Figure 1: Discrete Fourier Transform (DFT) visualizations of input observations, predicted time series, and ground truth between recent Transformer-based approaches and our proposed method.

or top-$K$ frequency selection, often capturing spurious correlations for forecasting (seen in Figure 1(c)).

In this paper, we propose FreCoformer to represent various short-term temporal variations in complex time series automatically. It is designed on top of the frequency domain and comprises three key designs: frequency patching operation and two independent observations of these patches. The *patching* operation refines the frequency bands, providing an opportunity to learn representations from detailed views of frequency components. The first observation, a *channel-wise attention mechanism*, weighs channel-wise correlations for each independent sub-frequency component. These independent attentions share model parameters across all sub-frequency learning, preventing winner-take-all of redundancy low-frequency components. The second observation is channel-independent, which summarizes global frequency information (i.e., *frequency-wise summarization*) and eliminates channel correlations to facilitate multivariate time series forecasting. We further propose a 'divide-and-conquer' forecasting framework that integrates FreCoformer with a long-term modeling module, deploying in the time domain, to improve the data fit of time series scenarios. Additionally, we present a lightweight variant of FreCoformer to alleviate computational load, extending our proposal to various large-scale datasets. Our main contributions lie in three folds.

**1)** FreCoformer is a novel forecasting module designed for computing frequency correlation for representing short-term variations in time series. It can automatically identify the relevant and consistent frequency components in historical observations and forecast data points. Figure 1(d) illustrates our superiority to different previous methodologies in complex datasets.

**2)** The divided-and-conquer framework enhances data fit, and the ablation study shows the distinct contributions of each module under varying data scenarios. Extensive experimental results on eight benchmarks show the effectiveness of our proposal, achieving superior performance, with 41 top-1 and 21 top-2 cases out of 64 in total.

**3)** We incorporate the Nyström approximation to reduce the computational complexity of attention maps, achieving lightweight models with competitive performance. This opens new possibilities for efficient time series forecasting. Interestingly, results demonstrate that Nyström-FreCoformer can particularly enhance performance in datasets with a large number of channels.

## 2 RELATED WORKS

**Transformer for Time Series Forecasting.** Forecasting is an important task in time series analysis (Alysha M. De Livera & Snyder, 2011; Hamilton, 2020). Transformer has recently achieved

a progressive breakthrough in time series forecasting (Nie et al., 2023; Zhang & Yan, 2023; Jiang et al., 2023). Earlier attempts make efforts to improve the computational efficiency of Transformers to form them for time series forecasting tasks (Beltagy et al., 2020; Zhou et al., 2021; Liu et al., 2022a). Several works further apply Transformers to the time domain of time series to model inherent temporal dependencies (Li et al., 2019; Zhou et al., 2021; Liu et al., 2022b; Nie et al., 2023). Various studies have integrated frequency decomposition and spectrum analysis with Transformer in modeling temporal variations (Wu et al., 2021; Woo et al., 2022b), to improve the capacity of temporal-spatial representation. The work of (Zhou et al., 2022) designs the attention layers that directly function in the frequency domain to enhance spatial or frequency representation.

**Modeling Short-term Variation in Time Series.** Short-term variations are intrinsic characteristics of time series data, playing a crucial role in effective forecasting (Crespo Cuaresma et al., 2004; Liu et al., 2022c). Numerous deep learning-based methods have been proposed to capture these transient patterns (Chung et al., 2014; Neil et al., 2016; Chang et al., 2018; Bai et al., 2018; Stoller et al., 2019; Wen et al., 2020; Wu et al., 2021; Woo et al., 2022a; Wang et al., 2022). Here, we summarize some works closely aligned with our proposal. Pyraformer ((Liu et al., 2022b) applies a pyramidal attention module with inter-scale and intra-scale connections to capture various temporal dependencies. FEDformer ((Zhou et al., 2022) incorporates the Fourier spectrum within the attention computation to identify pivotal frequency components. Beyond Transformers, TimesNet ((Wu et al., 2023) employs Inception blocks to capture both intra-period and inter-period variations.

**Channel-wise Correlation.** Understanding the cross-channel correlation is also critical for time series forecasting. Several studies aim to capture intra-channel temporal variations and subsequently model the inter-channel correlations using Graph Neural Networks (GNNs) (Wu et al., 2020; Cao et al., 2021). Recently, Crossformer (Zhang & Yan, 2023) proposes a two-stage attention layer designed to simultaneously capture temporal variations and their cross-channel correlations. Extensive experimental results have demonstrated its effectiveness in multivariate time series forecasting.

## 3 PROPOSED METHOD

Let $\mathbf{X} = \{\boldsymbol{x}_L^{(i)}\}_{m=1}^{C}$ denote a multivariate time-series consisting of $C$ channels, where each channel records an independent $L$ length historical observation. We aim to design an effective forecasting function $f_\theta(\cdot)$ that can accurately forecast $T$ data points for each channel, resulting in $\hat{\mathbf{X}} \in \mathbb{R}^{C \times T}$.

### 3.1 FreCoforme.

**Forward Process.** FreCoformer consists of four principal components: (1) a DFT-to-IDFT backbone, (2) frequency-wise patching, (3) channel-wise attention, and (4) frequency-wise summarization. An overview can be found in Figure 2(a). The DFT-to-IDFT backbone decomposes the input time series into its frequency components via DFT and learns a consistent representation of relevant frequency components (by frequency-wise patching, channel-wise attention, and frequency-wise summarization), enabling future time series generation through IDFT. Specifically,

(i) The input $\mathbf{X}$ is transformed to the real part $\mathbf{R} \in \mathbb{R}^{C \times F}$ and imaginary part $\mathbf{I} \in \mathbb{R}^{C \times F}$ of the frequency by DFT, where $F$ denotes the frequency bands.

(ii) Along the $C$-axis, we segment these two matrices into a sequence of $N$ sub-frequency patches, i.e., $(\mathbf{R}_1, ..., \mathbf{R}_N)$ and $(\mathbf{I}_1, ..., \mathbf{I}_N)$, for all channels to refine the frequency information.

(iii) Subsequently, cross-channel patches within the same sub-frequency are fed into the Transformer. Then, the Transformer sequentially and independently captures the channel-wise dependencies of each sub-frequency and, post-processing, concatenates all sub-frequencies.

(iv) Along the $F$-axis, we further abstract the overall frequency information, resulting in two new real $\hat{\mathbf{R}}$ and imaginary $\hat{\mathbf{I}}$ parts. These two parts serve as IDFT for forecasting $\hat{\mathbf{X}}$.

**Frequency-wise Patching.** Given the $\mathbf{R}$ and $\mathbf{I}$ matrices of DFT, a non-overlapping patching operation is performed on them. We segment the frequency entries of $(\boldsymbol{r}_1^{(m)}, ..., \boldsymbol{r}_F^{(m)})$ and $(\boldsymbol{i}_1^{(m)}, ..., \boldsymbol{i}_F^{(m)})$ of each channel into a set of sub-frequency patches with $P$ dimension, resulting in $(\hat{\boldsymbol{r}}_1^{(m)}, ..., \hat{\boldsymbol{r}}_N^{(m)})$ and $(\hat{\boldsymbol{i}}_N^{(m)}, ..., \hat{\boldsymbol{i}}_N^{(m)})$, where $N = F/P$ is the number of patches. Thus, the input $\mathbf{X}$ will result in:

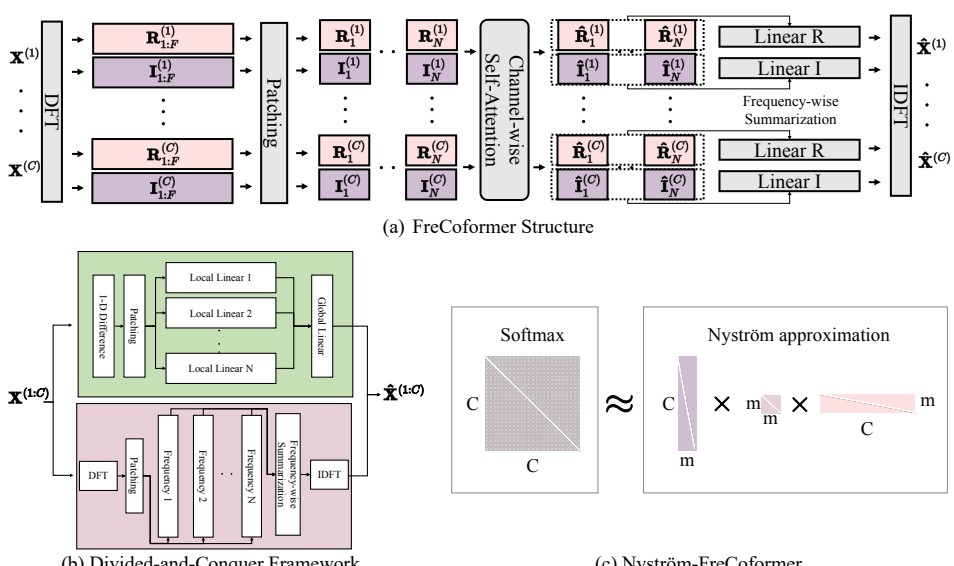

(a) FreCoformer Structure

(b) Divided-and-Conquer Framework

(c) Nyström-FreCoformer

Figure 2: System overview: (a) FreCoformer, (b) Divided-and-Conquer framework, and (c) Nyström-FreCoformer

$$(\mathbf{R}_1, ..., \mathbf{R}_N), (\mathbf{I}_1, ..., \mathbf{I}_N) = \text{Patching}(\text{DFT}(\mathbf{X})), \quad \mathbf{R}_{1:N}, \mathbf{I}_{1:N} \in \mathbb{R}^{C \times P}$$

The dimension of $P$ prevents information redundancy over fine-grained frequency bands, like neighboring 1Hz and 2Hz. This parameter is adjustable to real-world scenarios, e.g., an hourly sampling in daily recordings or alpha waveform typically occurring at 8–12 Hz (Adamantidis et al., 2019).

**Channel-wise Attention.** We employ the Transformer encoder to learn the frequency-independent channel-wise correlation. For the $n$-th sub-frequency, where $n \in 1, 2, ..., N$, $\hat{\boldsymbol{r}}_n^{(m)}$ and $\hat{\boldsymbol{i}}_n^{(m)}$ are concatenated as the embedding for each channel, yielding all channel patches $\mathbf{W}_n = \text{Concat}(\mathbf{R}_n, \mathbf{I}_n)$, where $\mathbf{W}_n \in \mathbb{R}^{C \times 2P}$. These patches are then mapped to the Transformer latent space of dimension $D$ via a linear projection $\mathbf{E}_n \in \mathbb{R}^{2P \times D}$, and a patch-wise normalization. This normalization is used to eliminate distributional differences across sub-frequency bands. Subsequently, we feed $C$ tokens of $\mathbf{W}_n' = \text{PreNorm}(\mathbf{W}_n \mathbf{E}_n)$, each at a time for self-attention computations, and this process is performed independently $N$ times for all sub-frequencies to obtain the complete representations. Therefore, the attention computation can be formalized as:

$$\mathbf{A}_n = \text{Attention}(\mathbf{Q}_n, \mathbf{K}_n, \mathbf{V}_n) = \text{Softmax}\left(\frac{\mathbf{W}_n' \mathbf{W}_n^q (\mathbf{W}_n' \mathbf{W}_n^k)^T}{\sqrt{d}}\right) \mathbf{W}_n' \mathbf{W}_n^v$$

where $\mathbf{W}_n^q, \mathbf{W}_n^k, \mathbf{W}_n^v \in \mathbb{R}^{D \times M}$ are the weight matrices for generating the query matrix $\mathbf{Q}_n$, key matrix $\mathbf{K}_n$, and value matrix $\mathbf{V}_n$. $\sqrt{d}$ denotes a scaling operation. The attention module also contains normalization and a feed-forward layer with residual connections (Dosovitskiy et al., 2021), and $\mathbf{A}_n \in \mathbb{R}^{C \times M}$ weights the correlations among $C$ channels for the $n$-th sub-frequency band.

**Frequency-wise Summarization.** We concatenate all independent attention maps $(\mathbf{A}_1, ..., \mathbf{A}_N)$, in sequence to form $\mathbf{A} \in \mathbb{R}^{C \times (N \times M)}$. Given $\mathbf{A}$ is derived from the $N$ times observation of independent frequency, we introduce a frequency-wise layer projection to summarize the overall frequency information, resulting in $\mathbf{A}'$. Ultimately, two distinct linear layers are employed to generate the refined real and imaginary parts, serving IDFT for forecasting.

$$\hat{\mathbf{X}} = \text{IDFT}(\hat{\mathbf{R}}, \hat{\mathbf{I}}), \quad \text{where } \hat{\mathbf{R}} = \text{Linear}(\mathbf{A}'); \hat{\mathbf{I}} = \text{Linear}(\mathbf{A}')$$

Notably, this frequency-wise summarization is channel-independent and shares the parameters of linear projection across all channels, i.e., $\mathbf{A}' = (\mathbf{A}'^{(1)}, ..., \mathbf{A}'^{(C)}) = \text{Linear}(\mathbf{A}^{(1)}, ..., \mathbf{A}^{(C)})$. This aims to mitigate channel correlations and enhance channel fits, referring to (Nie et al., 2023).

## 3.2 DIVIDED-AND-CONQUER FRAMEWORK

Real-world time series exhibit variability across different scenarios. For instance, analyzing long-term variations in the data can reflect seasonal-trend patterns, such as differences between summer and winter and weekly changes in air quality (Vito, 2016; Karevan & Suykens, 2020). Conversely, areas like banking transactions, electricity consumption, and hospital foot traffic (Crespo Cuaresma et al., 2004; Lai et al., 2018b; Alysha M. De Livera & Snyder, 2011) require a focus on short-term variations. A successful forecasting function should adapt to various scenarios and capture eventful patterns to ensure precise forecasting.

Therefore, we propose a 'divide-and-conquer' framework and introduce a simple linear projection-based module, incorporated into FreCoformer, to enhance adaptability to various types of time series data. Since FreCoformer is designed on top of the frequency domain, this new module, termed T-Net, operates in the time domain to further complement FreCoformer by improving the capability of modeling temporal dependencies.

Given the input $\mathbf{X} \in \mathbb{R}^{C \times T}$, the first-order difference operation is applied independently to each univariate time series to remove the non-stationary variations and noise, yielding $(\tilde{\mathbf{X}}^1, ..., \tilde{\mathbf{X}}^C)$. Drawing inspiration from the works of (Zeng et al., 2023; Nie et al., 2023), for the $m$-th series, we also segment the time domain of $\tilde{\mathbf{X}}^{(m)}$ into a sequence of $N'$ temporal patches, where $N' = L/P'$ denotes the number of patches, each of dimension $P'$. We form two-stage linear projections: initially capturing the local temporal dependencies of each patch and subsequently learning the global temporal dependencies after concatenating all the learned patches.

$$\hat{\mathbf{X}} = \text{Linear}^{\text{(global)}}(\text{Linear}^{\text{(local)}}(\tilde{\mathbf{X}}^1), ..., \text{Linear}^{\text{(local)}}(\tilde{\mathbf{X}}^C))$$

Notably, we intend for both FreCoformer and T-Net to independently learn different domain-based representations and each to have its own capacity to forecast the ground truth $\hat{\mathbf{X}}$. A summation is finally executed on the outputs of FreCoformer and T-Net without any additional operations.

## 3.3 NYSTRÖM-FRECOFORMER

The $O(n^2)$ memory and time complexity of self-attention is the bottleneck for using longer historical time series for forecasting (Li et al., 2019; Zhou et al., 2021; Nie et al., 2023). With the patching operations in both the time and frequency domains, $O(LC^2)$ complexity has reduced to $O(\frac{L}{P}C^2)$. However, due to the channel-wise attention of FreCoformer, the computational cost increases proportionally with the number of channels, potentially leading to computational overloads when a large number of channels. We hence propose a lightweight Frecoformer inspired by NyströmFormer (Xiong et al., 2021) and conduct a matrix approximation for the attention map. Two main motivations drive our approach: First, employing the

Table 1: Computation complexity. $L$ is the input sequence length, $C$ is the channel count, and $P$ denotes the patch dimension.

| Methods | Complexity |
|---|---|
| Fedformer | $O(LC)$ |
| PatchTST | $O(\frac{L^2}{s^2}C)$ |
| Crossfromer | $O(\frac{L^2}{P^2}C)$ |
| Ours | $O(\frac{L}{P}C^2)$ |
| Ours(Nyström) | $O(\frac{L}{P}C)$ |

Nyström matrix approximation method allows us to further reduce our complexity to $O(\frac{L}{S}C)$ without modifying the feature extraction (attention computation) or the data stream structure within the Transformer, as opposed to previous methods (Zhou et al., 2021; Liu et al., 2022a; Wu et al., 2021; Zhou et al., 2022). Second, real-world time series data often exhibit redundancy across different dimensions due to consistent characteristics among similar variables, like the traffic volumes of neighboring locations Zhou et al. (2021). This redundancy can lead to unnecessary correlation computations in channel-wise attention processes. To compute the attention matrix $\boldsymbol{A}$, we first select $m$ landmark columns from the input $\mathbf{Q}_n$ and $\mathbf{K}_n$ matrices in each channel, denoted as $\tilde{\mathbf{Q}}_n$ and $\tilde{\mathbf{Q}}_n$,

then compute:

$$\tilde{\mathbf{F}}_n = \text{softmax}(\frac{\mathbf{Q}_n\tilde{\mathbf{K}}_n^T}{\sqrt{d}}), \quad \tilde{\mathbf{A}}_n = \text{softmax}(\frac{\tilde{\mathbf{Q}}_n\tilde{\mathbf{K}}_n^T}{\sqrt{d}})^+, \quad \tilde{\mathbf{B}}_n = \text{softmax}(\frac{\tilde{\mathbf{Q}}_n\mathbf{K}_n^T}{\sqrt{d}})$$

Where $\tilde{\mathbf{A}}_n^+$ is the Moore-Penrose inverse of $\tilde{\mathbf{A}}_n$ (Xiong et al., 2021), and the Nyström approximation for $n$-th channel-wise attention $\mathbf{A}_n$ is:

$$\mathbf{A}_n \approx \tilde{\mathbf{A}}_n = \tilde{\mathbf{F}}_n\tilde{\mathbf{A}}_n\tilde{\mathbf{B}}_n.$$

With the use of Nyström approximation on attention maps, the computational load has reduced from $O(\frac{L}{P}C^2)$ to $O(\frac{L}{P}C)$. Detailed derivations and proofs can be found in Appendix A.1.

# 4 EXPERIMENTS

## 4.1 PROTOCOLS

Table 2: Benchmark datasets summary

| Datasets | Weather | Electricity | ETTh1 | ETTh2 | ETTm1 | ETTm2 | Air | Traffic |
|---|---|---|---|---|---|---|---|---|
| #channel | 21 | 321 | 7 | 7 | 7 | 7 | 12 | 862 |
| #timesteps | 52969 | 26304 | 17420 | 17420 | 69680 | 69680 | 6941 | 17544 |

**Datasets.** We conducted extensive experiments on eight real-world benchmark datasets: Weather, four ETT datasets (ETTh1, ETTh2, ETTm1, ETTm2), Electricity, Traffic, and Air [1] (Vito, 2016), where the former seven datasets are available in the work (Wu et al., 2021) [2]. A summary of the datasets is presented in Table 2 and details can be found in Appendix A.2.

**Baselines.** We selected some state-of-the-art (SOTA) time series forecasting works as our baselines: PatchTST (Nie et al., 2023), TimesNet (Wu et al., 2023), Fedformer (Zhou et al., 2022) and Pyraformer (Liu et al., 2022a). PatchTST represents a new SOTA, outperforming several authoritative early works, including Autoformer (Wu et al., 2021), Informer (Zhou et al., 2021), and DLinear (Zeng et al., 2023). Other baselines with differing architectures are designed to capture short-term temporal variations. Besides, we compared our proposal to a multi-channel modeling SOTA, i.e., Crossformer (Zhang & Yan, 2023).

**Setup.** All baselines adhere to the same prediction length with $T \in \{24, 36, 48, 60\}$ for the Air dataset and $T \in \{96, 192, 336, 720\}$ for other datasets. The look-back window $L = 336$ was used in our setting for fair comparisons, referring to (Nie et al., 2023). Besides, we further explored the impact of an extended look-back window by evaluating with $L = 512$.
– For the Air dataset, we tested our model and all baselines with a look-back window $L = 104$, based on the settings recommended for small datasets in (Zhou et al., 2021).
– For other datasets, we collected all available results of PatchTST, Fedformer, and Pyraformer from (Nie et al., 2023). Results for Crossformer with prediction lengths $T \in \{336, 720\}$ were collected from (Zhang & Yan, 2023). For unavailable $T \in \{96, 192\}$, we implemented Crossformer to obtain the results. We collected results of TimesNet from (Wu et al., 2023) with the default $L = 96$ and implemented TimesNet with our default $L = 336$ to select the best outcomes for a fair comparison.

## 4.2 RESULTS

### 4.2.1 MAIN RESULTS

Table 3 shows the main results for multivariate long-term forecasting. Overall, with default a look-back window of $L = 336$, our proposal shows leading performance on most datasets, as well as on different prediction length settings, with 27 top-1 and 34 top-2 cases out of 64 in total. When the look-back window is extended to $L = 512$, our framework demonstrates superior performance, achieving 41 top-1 and 21 top-2 rankings out of 64 cases. Considering both look-back Twindow settings, our framework achieves top-1 rankings in 63 out of 64 cases.

---

[1] https://archive.ics.uci.edu/dataset/360/air+quality
[2] https://drive.google.com/drive/folders/1ZOYpTUa82_jCcxIdTmyr0LXQfvaM9vIy

Table 3: Multivariate long-term forecasting results with MSE/MAE. Bold/underline indicates the best/second results. The asterisk* denotes the results are implemented by us; Other results are from original papers (Nie et al., 2023; Zhang & Yan, 2023; Wu et al., 2023).

| Models | | Ours 512 | | Ours 336 | | PatchTST | | Crossformer | | TimesNet | | Fedformer | | Pyraformer | |
|---|---|---|---|---|---|---|---|---|---|---|---|---|---|---|---|---|
| Metric | | MSE | MAE | MSE | MAE | MSE | MAE | MSE | MAE | MSE | MAE | MSE | MAE | MSE | MAE |
| Weather | 96 | **0.146** | 0.194 | 0.149 | **0.196** | 0.152 | 0.199 | 0.166* | 0.238* | 0.167* | 0.227 | 0.238 | 0.314 | 0.896 | 0.556 |
| | 192 | **0.190** | 0.240 | 0.193 | **0.238** | 0.197 | 0.243 | 0.232* | 0.309* | 0.214 | 0.263 | 0.275 | 0.329 | 0.622 | 0.624 |
| | 336 | **0.242** | **0.277** | 0.245 | 0.279 | 0.249 | 0.283 | 0.266* | 0.326* | 0.269* | 0.301 | 0.339 | 0.377 | 0.739 | 0.753 |
| | 720 | **0.315** | **0.334** | 0.318 | 0.332 | 0.320 | 0.335 | 0.353* | 0.393* | 0.341 | 0.350 | 0.389 | 0.409 | 1.004 | 0.934 |
| Electricity | 96 | **0.128** | 0.224 | 0.129 | 0.225 | 0.130 | **0.222** | 0.198* | 0.289* | 0.168 | 0.272 | 0.186 | 0.302 | 0.386 | 0.449 |
| | 192 | **0.145** | **0.239** | 0.146 | 0.240 | 0.148 | 0.240 | 0.239* | 0.315* | 0.184 | 0.289 | 0.197 | 0.311 | 0.386 | 0.443 |
| | 336 | **0.162** | **0.259** | 0.165 | 0.259 | 0.167 | 0.261 | 0.404 | 0.423 | 0.198 | 0.300 | 0.213 | 0.328 | 0.378 | 0.443 |
| | 720 | **0.197** | **0.290** | 0.202 | 0.294 | 0.202 | 0.291 | 0.433 | 0.438 | 0.220 | 0.320 | 0.233 | 0.344 | 0.376 | 0.445 |
| ETTh1 | 96 | **0.359** | **0.390** | 0.362 | 0.391 | 0.375 | 0.399 | 0.424* | 0.444* | 0.384 | 0.402 | 0.376 | 0.415 | 0.664 | 0.612 |
| | 192 | **0.390** | **0.408** | 0.403 | 0.411 | 0.414 | 0.421 | 0.602* | 0.555* | 0.436 | 0.429 | 0.423 | 0.446 | 0.790 | 0.681 |
| | 336 | **0.401** | **0.418** | 0.406 | 0.415 | 0.431 | 0.436 | 0.440 | 0.461 | 0.491 | 0.469 | 0.444 | 0.462 | 0.891 | 0.738 |
| | 720 | 0.436 | 0.459 | **0.433** | **0.452** | 0.449 | 0.466 | 0.519 | 0.524 | 0.515* | 0.498* | 0.469 | 0.492 | 0.963 | 0.782 |
| ETTh2 | 96 | **0.268** | **0.339** | 0.273 | 0.335 | 0.274 | 0.336 | 0.801* | 0.635* | 0.340 | 0.374 | 0.332 | 0.374 | 0.645 | 0.597 |
| | 192 | **0.328** | **0.373** | 0.337 | 0.378 | 0.339 | 0.379 | 0.854* | 0.665* | 0.402 | 0.414 | 0.407 | 0.446 | 0.788 | 0.683 |
| | 336 | **0.322** | **0.379** | 0.323 | 0.379 | 0.331 | 0.380 | 0.943* | 0.755* | 0.452 | 0.452 | 0.400 | 0.447 | 0.907 | 0.747 |
| | 720 | **0.372** | **0.421** | 0.374 | 0.419 | 0.379 | 0.422 | 1.146* | 0.814* | 0.462 | 0.468 | 0.412 | 0.469 | 0.963 | 0.782 |
| ETTm1 | 96 | 0.286 | 0.340 | **0.285** | **0.338** | 0.290 | 0.342 | 0.378* | 0.371* | 0.338 | 0.375 | 0.326 | 0.390 | 0.543 | 0.510 |
| | 192 | 0.326 | 0.366 | **0.322** | 0.362 | 0.332 | 0.369 | 0.394* | 0.435* | 0.374 | 0.362 | 0.365 | 0.415 | 0.557 | 0.537 |
| | 336 | 0.356 | **0.384** | 0.353 | 0.385 | 0.366 | 0.392 | 0.404 | 0.427 | 0.410 | 0.411 | 0.392 | 0.425 | 0.754 | 0.655 |
| | 720 | 0.416 | 0.424 | **0.409** | **0.420** | 0.420 | 0.424 | 0.569 | 0.528 | 0.478 | 0.450 | 0.446 | 0.458 | 0.908 | 0.724 |
| ETTm2 | 96 | 0.165 | 0.257 | **0.164** | **0.254** | 0.165 | 0.255 | 0.371* | 0.427* | 0.187 | 0.267 | 0.180 | 0.271 | 0.435 | 0.507 |
| | 192 | 0.220 | 0.295 | **0.218** | **0.291** | 0.220 | 0.292 | 0.553* | 0.535* | 0.249 | 0.309 | 0.252 | 0.318 | 0.730 | 0.673 |
| | 336 | **0.270** | 0.327 | 0.270 | **0.326** | 0.278 | 0.329 | 1.556* | 0.906* | 0.321 | 0.351 | 0.324 | 0.364 | 1.201 | 0.845 |
| | 720 | **0.358** | 0.383 | 0.361 | **0.381** | 0.367 | 0.385 | 1.566* | 0.984* | 0.405* | 0.399* | 0.410 | 0.420 | 3.625 | 1.451 |
| Air* | 24 | 0.577 | 0.568 | **0.572** | **0.562** | 0.607* | 0.582* | 0.574* | 0.581 | 0.642* | 0.602* | 0.646* | 0.602* | 0.717* | 0.661* |
| | 36 | **0.665** | 0.615 | 0.665 | **0.612** | 0.683* | 0.621* | 0.763* | 0.695* | 0.746* | 0.653* | 0.758* | 0.654* | 0.763* | 0.685* |
| | 48 | 0.702 | 0.634 | **0.695** | **0.630** | 0.722* | 0.644* | 0.888* | 0.735* | 0.766* | 0.662* | 0.854* | 0.697* | 0.762* | 0.687 |
| | 60 | **0.720** | **0.645** | 0.738 | 0.653 | 0.766* | 0.667* | 0.815* | 0.721* | 0.809* | 0.685* | 0.904* | 0.720* | 0.916* | 0.759* |
| Traffic | 96 | **0.356** | **0.248** | 0.358 | 0.250 | 0.367 | 0.251 | 0.502* | 0.299* | 0.593 | 0.321 | 0.576 | 0.539 | 2.085 | 0.468 |
| | 192 | **0.378** | **0.257** | 0.378 | 0.259 | 0.385 | 0.259 | 0.507* | 0.287* | 0.617 | 0.336 | 0.610 | 0.380 | 0.867 | 0.467 |
| | 336 | **0.384** | **0.260** | 0.391 | 0.264 | 0.398 | 0.265 | 0.513 | 0.289 | 0.629 | 0.336 | 0.608 | 0.375 | 0.869 | 0.469 |
| | 720 | **0.424** | **0.283** | 0.426 | 0.285 | 0.434 | 0.287 | 0.530 | 0.300 | 0.640 | 0.350 | 0.621 | 0.375 | 0.881 | 0.473 |

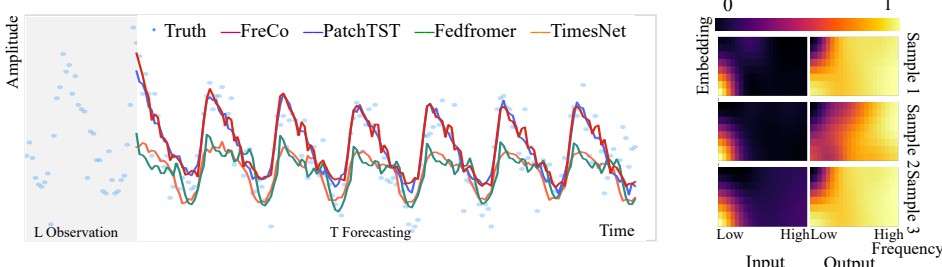

(a) Illustration of the prediction results on the ETTh1 dataset     (b) Illustration of the frequency features

Figure 3: (a)Visualized predictions from our model and baselines on the ETTh1 dataset. The X-axis denotes time steps, Y-axis is the amplitude of the time series. (b) Heatmaps of the input and output matrixes of FreCoformer's Transformer encoder on ETTh1, we showed 3 samples from different channels. These output matrixes will be used to generate forecasting. The X-axis denotes frequency components, Y-axis is the dimension of the feature vector. These heat maps show the energy distribution in the frequency domain.

### 4.2.2 MODEL ANALYSIS

Figure 1 already illustrates the ability of our proposal to accurately capture mid-to-high frequency components, demonstrating superiority over time-domain modeling methods (PatchTST), frequency decomposition-assisted temporal modeling methods (Autoformer), and frequency attention methods (Fedformer). We further visualize the time domain representation with more advanced baselines in

Table 4: Left part: Module ablation of our framework, FreCoformer only, and T-Net only, where bold/underline indicates the best/second results. Right part: Ablation study of channel-wise attention and frequency patching, where * denotes the better forecasting performance. 'Non-CW' denotes the removal of channel attention, replaced by an alternative linear projection; 'Non-FP' indicates that the entire frequency bands are used as tokens for channel-wise attention.

| Setting | | Complete | | FreCoformer | | T-Net | | Non-CW | | Non-FP | |
|---|---|---|---|---|---|---|---|---|---|---|---|
| Dataset | Metric | MSE | MAE | MSE | MAE | MSE | MAE | MSE | MAE | MSE | MAE |
| ETTh1 | 96 | **0.362** | **0.391** | 0.364 | 0.391 | 0.371 | 0.399 | 0.372* | 0.398* | 0.373 | 0.400 |
| | 192 | **0.403** | **0.411** | **0.403** | 0.412 | 0.411 | 0.421 | 0.405* | 0.414* | 0.410 | 0.419 |
| | 336 | **0.406** | **0.415** | 0.416 | 0.423 | 0.420 | 0.439 | 0.419* | 0.424* | 0.423 | 0.429 |
| | 720 | **0.433** | **0.452** | 0.434 | **0.452** | 0.446 | 0.464 | 0.435* | 0.453* | 0.458 | 0.469 |
| Weather | 96 | **0.149** | **0.196** | 0.173 | 0.225 | 0.150 | 0.197 | 0.176 | 0.227 | 0.174* | 0.225* |
| | 192 | **0.193** | **0.238** | 0.216 | 0.262 | 0.194 | 0.239 | 0.218 | 0.262* | 0.217* | 0.262* |
| | 336 | **0.245** | **0.279** | 0.263 | 0.295 | 0.246 | 0.280 | 0.265* | 0.295* | 0.266 | 0.298 |
| | 720 | **0.318** | **0.332** | 0.328 | 0.342 | 0.319 | 0.333 | 0.332* | 0.343* | 0.333 | 0.347 |

Figure 3(a). Both input and output are from the ETTh1 dataset, and the length is 336. Fedformer and TimesNet fail to accurately capture both long-term and short-term patterns. Compared to the best-performing PatchTST, our model exhibits an advantage in identifying short-term variations, resulting in detailed fluctuations in periodicity variation. More results can be seen in Appendix A.3.

To demonstrate the efficacy of the core design—channel-wise attention in FreCoformer, we visualized the heatmaps of the input and output DFT matrices of the Transformer encoder in FreCoformer in Figure 3(b). The energy of the original data is primarily concentrated in the low-frequency range, leading to a potential imbalance in energy distribution. In the output of the transformer encoder, there is a balanced energy distribution between low-frequency and mid-to-high-frequency components. This balance likely enables our method to efficiently extract pivotal frequency features across the entire frequency spectrum and various temporal variations, enhancing prediction outcomes.

### 4.2.3 ABLATION STUDY

**Module Ablation Study.** We investigate the efficiency of our framework and its modules by using the ETTh1 and Weather datasets. The ETTh1 dataset contains more intricate mid-to-high-frequency information, while the Weather dataset primarily focuses on low-frequency data. We independently implement two modules for forecasting these datasets and compare their results to the complete framework (in Table 4 (Left)). It shows on datasets like ETTh1, which are rich in complex high-frequency information, FreCoformer consistently has better performance. Conversely, on datasets like Weather, where long-term variations (low frequency) are dominant, using solely the time domain modeling has better outcomes, but combining both has superior results. These observations imply that frequency modeling has more contributions to our framework in intricate datasets. Also, it does not bring redundancy to time domain modeling in simple and stationary time series.

**Channel-wise Attention and Frequency Patching Ablations** We further investigate the impact of channel-wise attention and frequency patching (refinement) on forecasting accuracy. As shown in Table 4 (Right), our framework consistently achieved superior accuracy in all experiments. In datasets like ETTh1, characterized by more complex frequency information, channel-wise attention achieves better performance in forecasting than frequency patching, emphasizing the significance of our fundamental design of channel-wise attention.

### 4.2.4 NYSTRÖM-FRECOFORMER

We conduct comparative experiments to evaluate forecasting accuracy and computational complexity against various baseline methods. In Figure 4, the X-axis denotes GPU memory usage, while the Y-axis indicates prediction accuracy. Obviously, our framework outperforms in terms of accuracy across both datasets. In datasets with fewer channels, like ETTh1 (7 channels), our model excels in both accuracy and computational efficiency. In contrast, when dealing with datasets having a larger number of channels, like the Weather (21 channels), our original method still retains

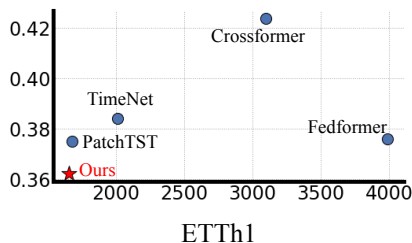 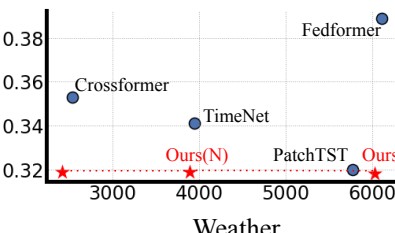

Figure 4: Visualization of prediction accuracy and computational complexity comparing various baselines, FreCoformer, and Nyström-FreCoformer.

the highest accuracy, though with a slight increase in computational load (as denoted by 'Ours' in Figure 4 (Right)). We further maintain constant parameters, modifying only the computational method for self-attention by employing Nyström, which allowed for a substantial reduction in computational demand without sacrificing accuracy (Ours(N)). Moreover, refining the parameters in the Nyström variant allowed us to realize further computational efficiencies without compromising accuracy (Nyström-freCoformer). Consequently, our model demonstrates superiority in both computational cost and accuracy in this setup.

Table 5: A more comprehensive comparison, taking into account MSE and GPU memory usage. We used the ETTh1 and Weather, Electricity, Traffic dataset with a look-back window of length 336. For ETTh1, the prediction length is 96, and for Weather, Electricity, and Traffic, it's 720 time steps. We evaluated the effectiveness of various methods based on these metrics and also considered runtime memory consumption. The best results are in **bold** and the second best results are in underlined.

| Dataset(channels) | Metric | Ours | Ours (Nyström) | PatchTST | Crossfromer | TimesNet | Fedformer |
|---|---|---|---|---|---|---|---|
| ETTh1(7) | MSE | **0.362** | - | 0.375 | 0.424 | 0.384 | 0.376 |
| | $O$ | **1661** | - | 1683 | 3096 | 2015 | 3989 |
| Weather(21) | MSE | **0.318** | 0.319 | 0.320 | 0.353 | 0.341 | 0.389 |
| | $O$ | 6029 | **2422** | 5775 | 2635 | 3943 | 6115 |
| Electricity(321) | MSE | **0.129** | 0.129 | 0.130 | 0.198 | 0.168 | 0.186 |
| | $O$ | 10113 | **6261** | 17379 | 15375 | 15593 | 7285 |
| Traffic(862) | MSE | 0.431 | **0.426** | 0.434 | 0.530 | 0.640 | 0.621 |
| | $O$ | 22317 | **13357** | 33823 | 39387 | 25689 | 17023 |

## 5 CONCLUSION

This paper proposes an effective design of Transformer-based models for modeling short-term temporal variation through frequency modeling, termed FreCoformer. FreCoformer is built upon the Transformer model and has three key components: (i) frequency refinement, (ii) channel-wise attention to independent frequency bands, and (iii) frequency-wise summarization. Compared to the previous works, FreCoformer locally and globally learns the frequency correlations of various short-term variations in time series. We further propose a divide-and-conquer framework and introduce a simple linear projection-based module incorporated into FreCoformer, to enhance adaptability to various types of time series data. Extensive experiments show the effectiveness of our proposal can outperform other baselines in different real-world time series datasets. The ablation shows the success of our FreCoformer design. We further incorporate Nyström approximation to reduce the computational complexity of attention maps, achieving lightweight with competitive forecasting performance. This introduces a new perspective for effective time series forecasting. Interestingly, results show that Nyström-FreCoformer can further enhance model performance in the time series data with a large number of channels.

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
