

Figure 1: The actual values on two different datasets, the prediction results from T-Net, the prediction results from FreCoformer, and the final prediction results of the combined model.

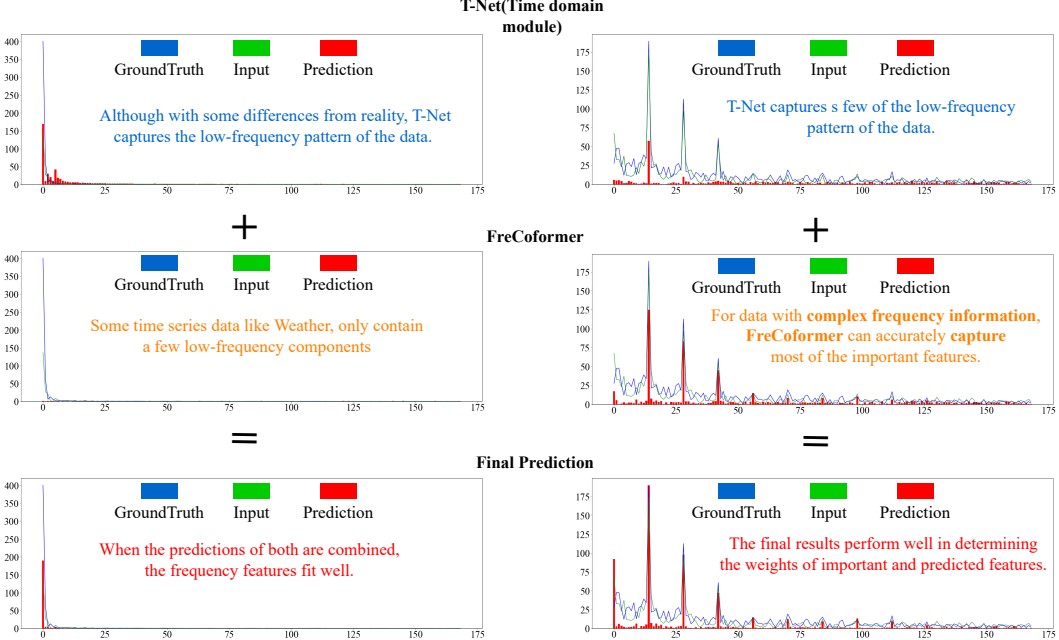

Figure 2: Each figure consists of visualizations of the frequency domain features for input data, actual values, and prediction results. On the left side is the weather dataset, and on the right side is the ETTh1 dataset. There are three figures on each side, with the top, middle, and bottom rows representing the prediction results from T-Net, the prediction results from FreCoformer, and the final prediction results, respectively.

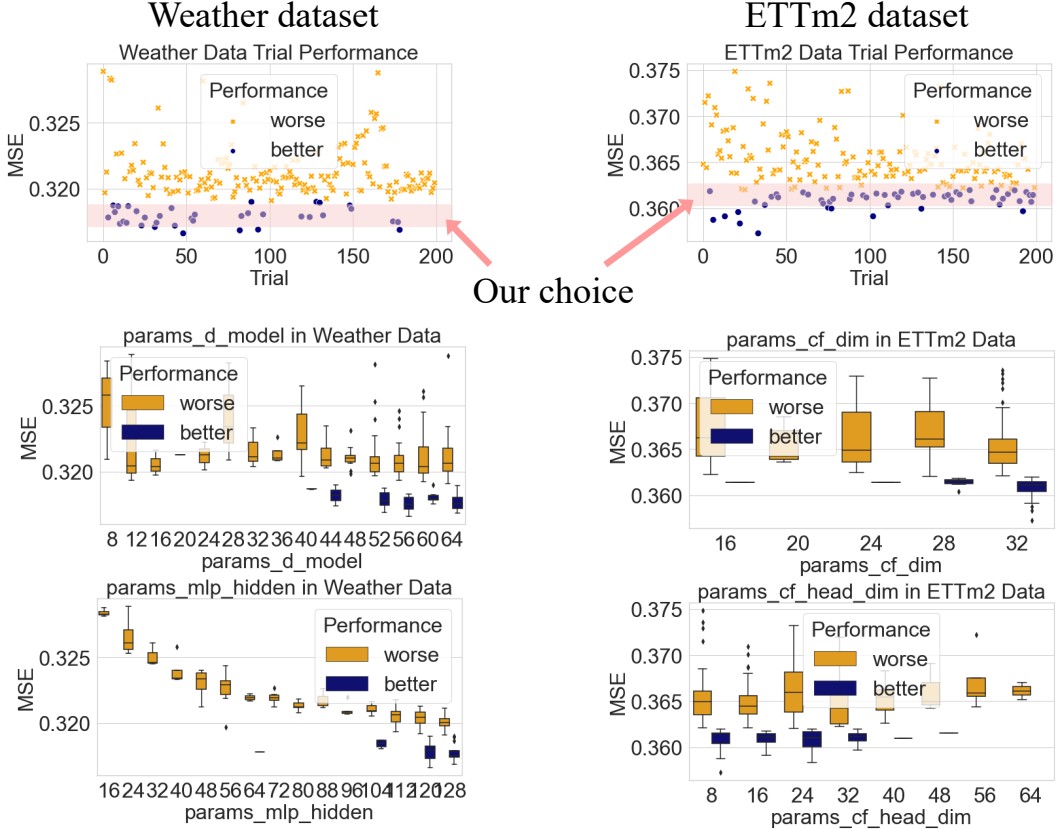

Figure 3: We selected the Weather dataset (mainly containing low-frequency features) and the ETTm2 data set (containing high-frequency features) to conduct 200 random settings of parameter experiments for various parameter combinations and visually displayed the results. Then, for the two data sets, the two parameters with the greatest influence were selected for visual display in box plots.

It can be seen that for the Weather data set, the two parameters with the greatest impact are the parameters of T-Net, while for ETTm2, the two parameters with the greatest impact are the two parameters of the FreCoformer. This is in line with our preset, that is, facing different types of data, two different components in the framework exert different effects.

"Better" here means that the result accuracy is higher than what we show in the paper and "worse" means that the result accuracy is lower. Although some of the results are better than the results we showed in the paper, these parameters with the highest accuracy were not used in the paper because we selected a parameter combination that is more stable on average.

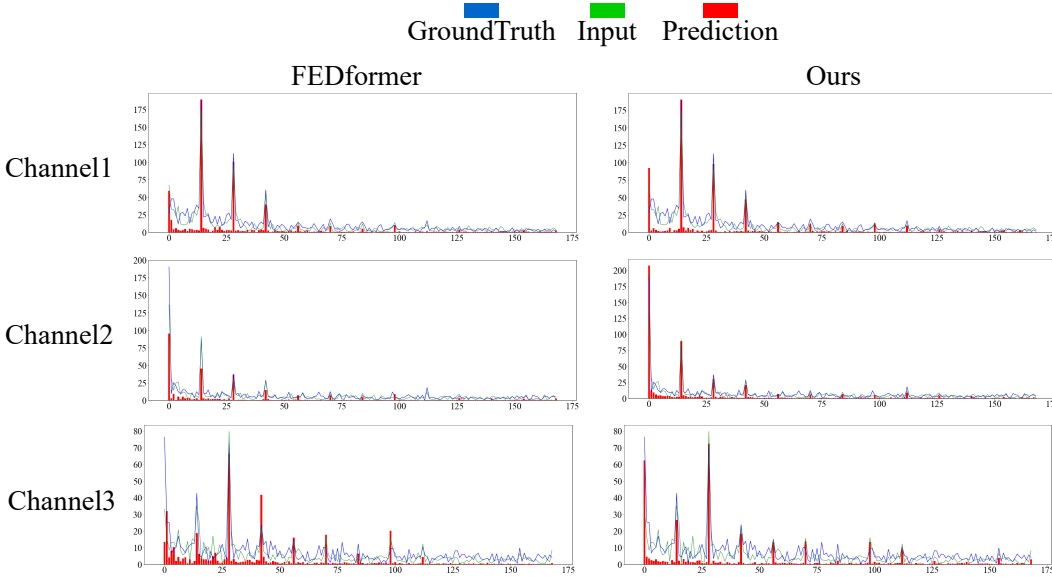

Figure 4: We select 3 different channels from ETTh1 data. The left part is FEDformer's result, and the right part is ours.

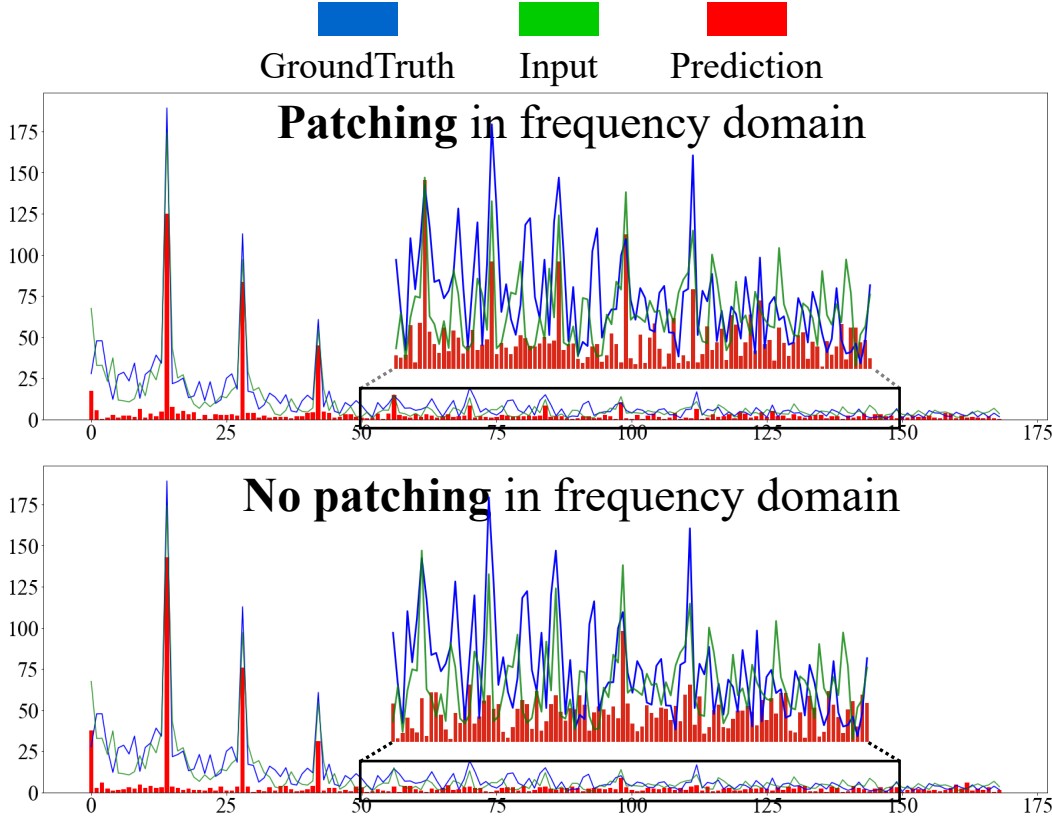

Figure 5: For the comparison regarding whether patching is used in the frequency domain, the top part represents the normal usage of patching, while the bottom part represents the scenario where patching is not used, and the entire frequency domain information is processed directly.

# A APPENDIX

## A.1 NYSTRÖM APPROXIMATION IN TRANSFORMER'S SELF-ATTENTION MECHANISM

We reduce the computational cost of self-attention in FreCofroemr's Transformer encoder using the Nyström method. Following, we describe how to use the Nyström method to approximate the softmax matrix in self-attention by sampling a subset of columns and rows.

Consider the softmax matrix in self-attention, defined as:

$$S = \text{softmax}\left(\frac{QK^T}{\sqrt{d_q}}\right)$$

This matrix can be partitioned as:

$$S = \begin{bmatrix} A_S & B_S \\ F_S & C_S \end{bmatrix}$$

Where $A_S$ is derived by sampling $m$ columns and rows from $S$.

By employing the Nyström method, the SVD of $A_S$ is given by:

$$A_S = U\Lambda V^T$$

Using this, an approximation $\hat{S}$ of $S$ can be constructed:

$$\hat{S} = \begin{bmatrix} A_S & B_S \\ F_S & F_S A_S^+ B_S \end{bmatrix}$$

Where $A_S^+$ is the Moore-Penrose inverse of $A_S$.

To further elaborate on the approximation, given a query $q_i$ and a key $k_j$, let:

$$\mathcal{K}(q_i, K) = \text{softmax}\left(\frac{q_i K^T}{\sqrt{d_q}}\right)$$

$$\mathcal{K}(Q, k_j) = \text{softmax}\left(\frac{Q k_j^T}{\sqrt{d_q}}\right)$$

From the above, we can derive:

$$\phi(q_i, K) = \Lambda^{-\frac{1}{2}} V^T \mathcal{K}(q_i, K)_{m \times 1}$$

$$\phi(Q, k_j) = \Lambda^{-\frac{1}{2}} U^T \mathcal{K}(Q, k_j)_{m \times 1}$$

Thus, the Nyström approximation for a particular entry in $\hat{S}$ is:

$$\hat{S}_{ij} = \phi(q_i, K)^T \phi(Q, k_j)$$

In matrix form, $\hat{S}$ can be represented as:

$$\hat{S} = \text{softmax}\left(\frac{QK^T}{\sqrt{d_q}}\right)_{n \times m} A_S^+ \text{softmax}\left(\frac{QK^T}{\sqrt{d_q}}\right)_{m \times n}$$

This method allows for the approximation of the softmax matrix in self-attention, potentially offering computational benefits.

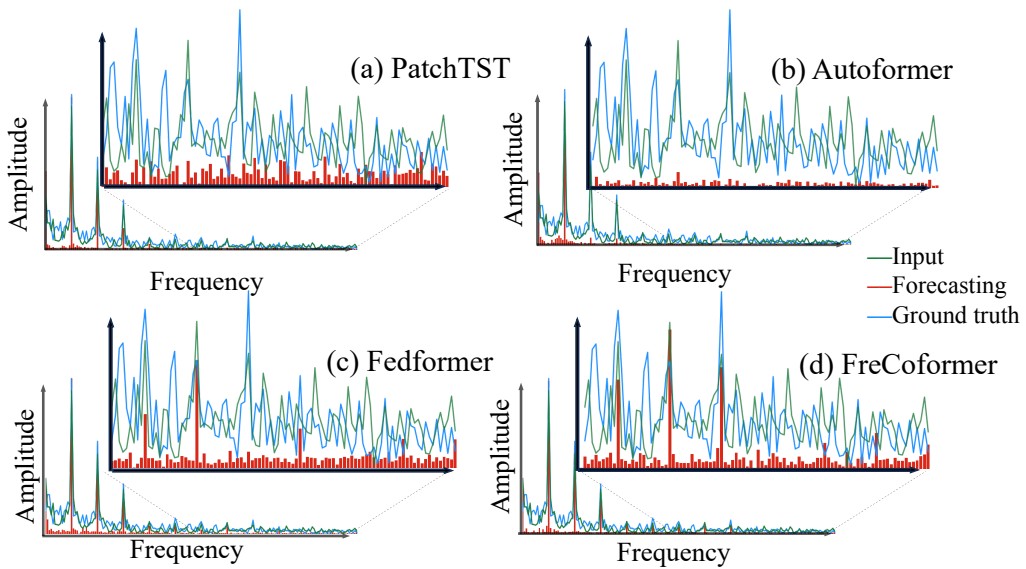

Figure 5: The visualization of frequency features in the prediction results is as follows: The horizontal axis represents frequency components, and the vertical axis represents the amplitude. It can be observed that the model's ability to extract frequency features progressively decreases, especially in the middle and high-frequency features. The dataset is ETTh1.

## A.2 DETAILS OF THE DATASETS

Weather contains 21 channels (e.g., temperature and humidity) and is recorded every 10 minutes in 2020. ETT (Zhou et al., 2021) (Electricity Transformer Temperature) consists of two hourly-level datasets (ETTh1, ETTh2) and two 15-minute-level datasets (ETTm1, ETTm2). Air Quality UCI (Vito, 2016) consists of 6,941 hourly average response instances of a chemical sensor array containing 12 different indicators embedded in air quality monitoring devices, specifically metal oxide chemical sensors. (We have already removed entries containing missing values.) The dataset contains 9358 instances of hourly averaged responses from an array of 5 metal oxide chemical sensors embedded in an Air Quality Chemical Multisensor Device. Electricity (Lai et al., 2018a), from the UCI Machine Learning Repository and preprocessed by, is composed of the hourly electricity consumption of 321 clients in kWh from 2012 to 2014. Traffic contains hourly road occupancy rates measured by 862 sensors on San Francisco Bay area freeways from January 2015 to December 2016.

We divided the eight datasets into two categories based on whether they exhibited complex mid-to-high-frequency features. The first category included datasets with more complex frequency features, such as Electricity, ETTh1, ETTh2, and Traffic. The second category consisted of datasets with relatively simple frequency features, including Weather, ETTm1, ETTm2, and Air.

## A.3 MORE RESULTS

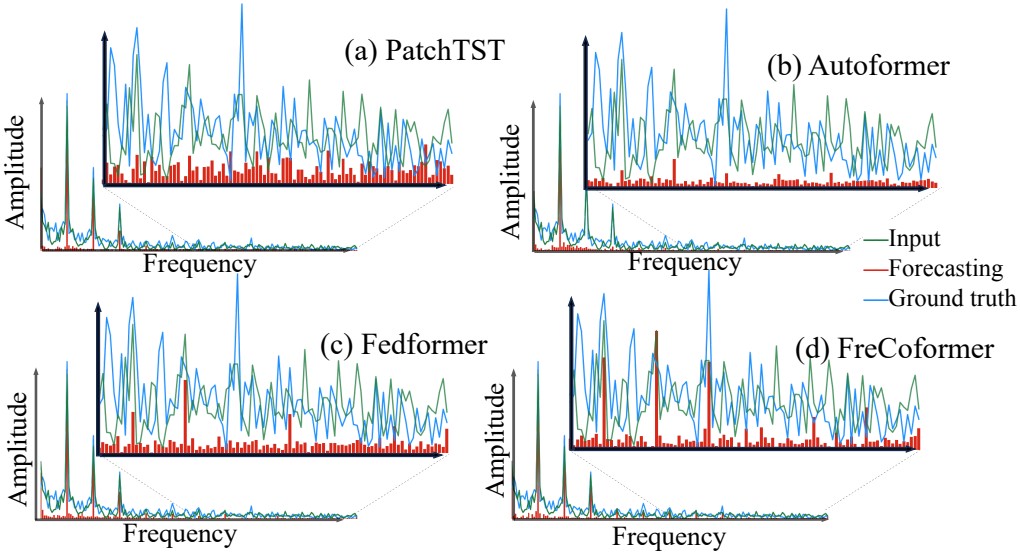

Figure 6: Another visualization of frequency features in the prediction results is as follows: The horizontal axis represents frequency components, and the vertical axis represents the amplitude. It can be observed that the model's ability to extract frequency features progressively decreases, especially in the middle and high-frequency features. The dataset is ETTh1.

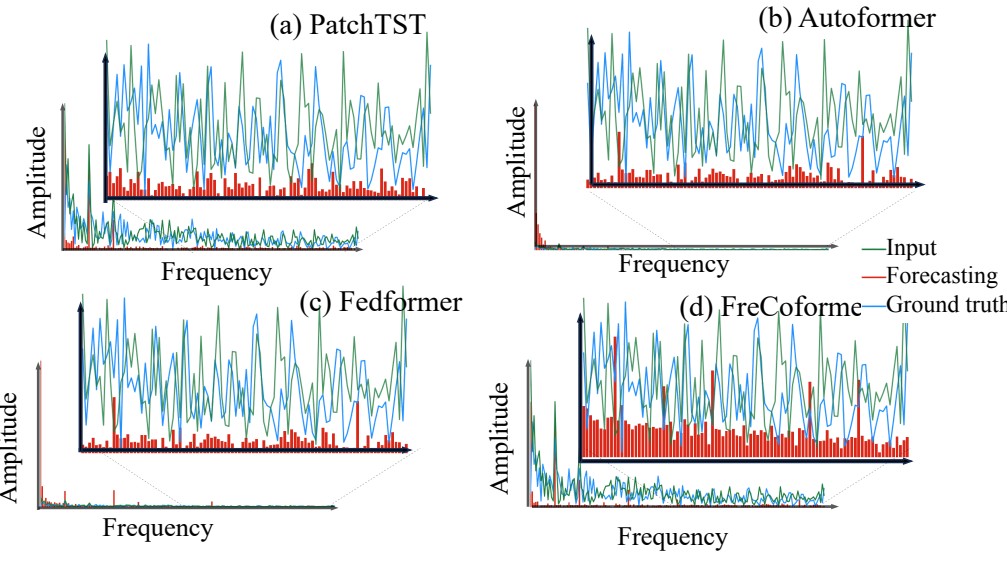

Figure 7: The visualization of frequency features in the prediction results is as follows: The horizontal axis represents frequency components, and the vertical axis represents the amplitude. It can be observed that the model's ability to extract frequency features progressively decreases, especially in the middle and high-frequency features. The dataset is ETTh2.

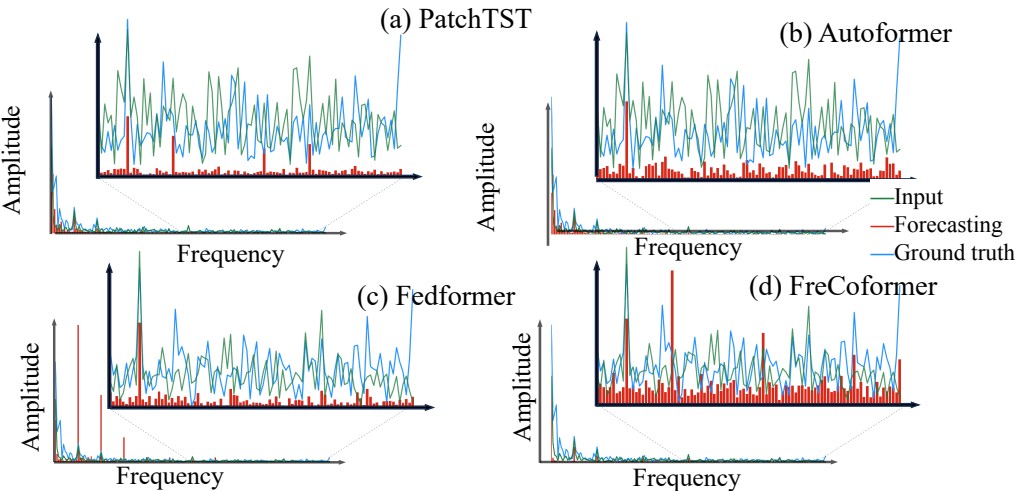

Figure 8: Another visualization of frequency features in the prediction results is as follows: The horizontal axis represents frequency components, and the vertical axis represents the amplitude. It can be observed that the model's ability to extract frequency features progressively decreases, especially in the middle and high-frequency features. The dataset is ETTh2.

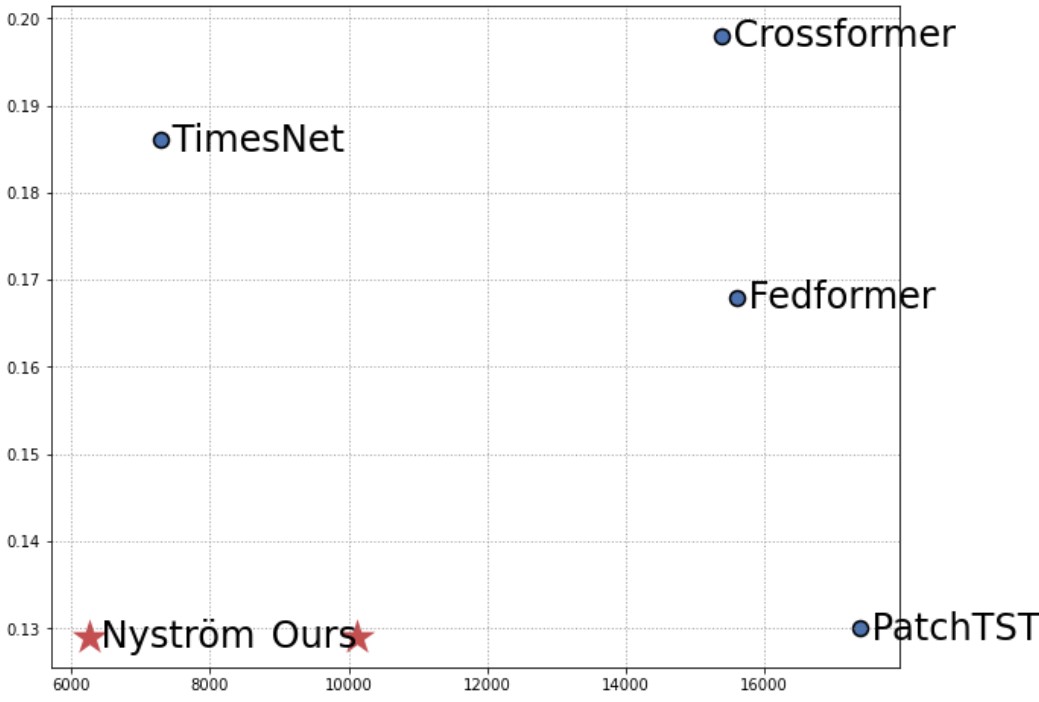

Figure 9: The prediction accuracy and computational complexity on the Electricity dataset.

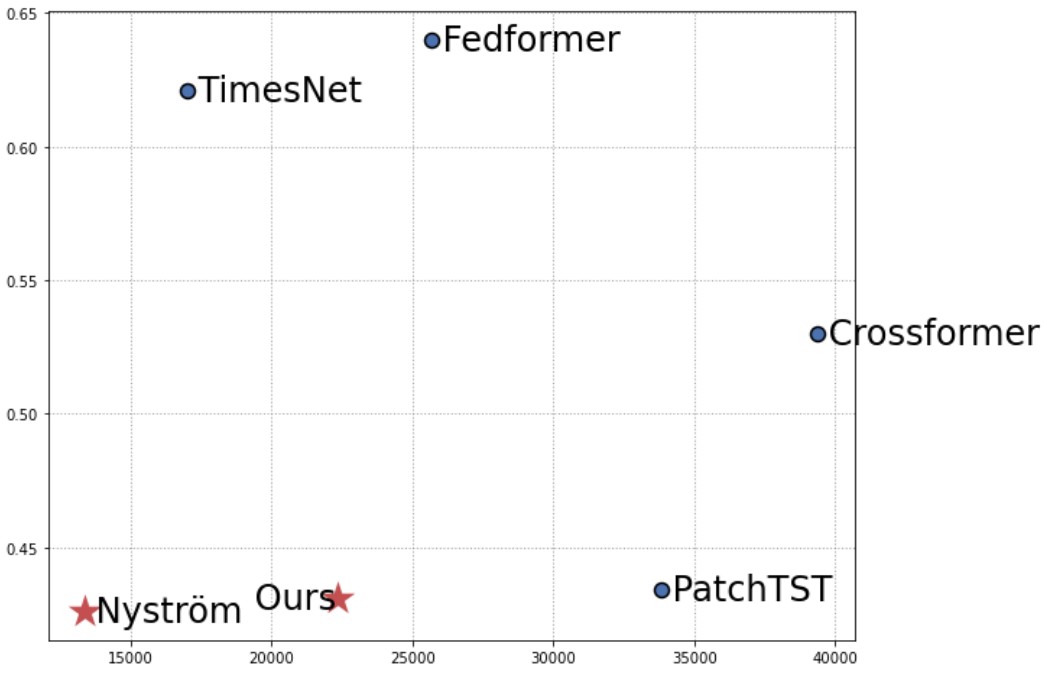

Figure 10: The prediction accuracy and computational complexity on the Traffic dataset.

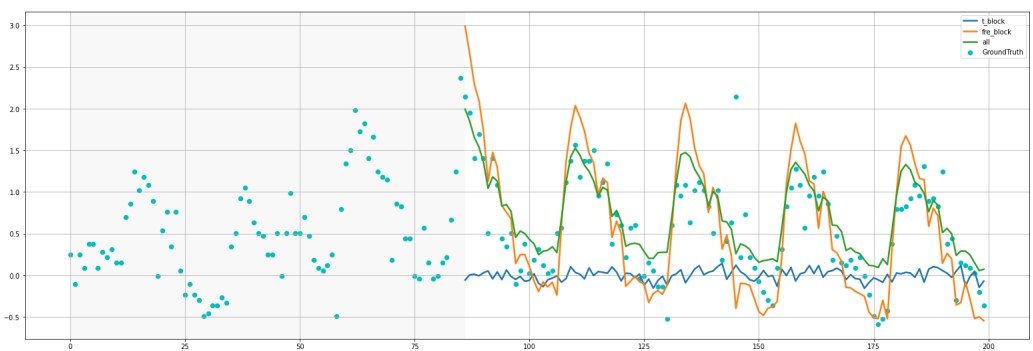

Figure 11: We visualized the outputs of T-Net, FreCoformer, and the final fused output separately.

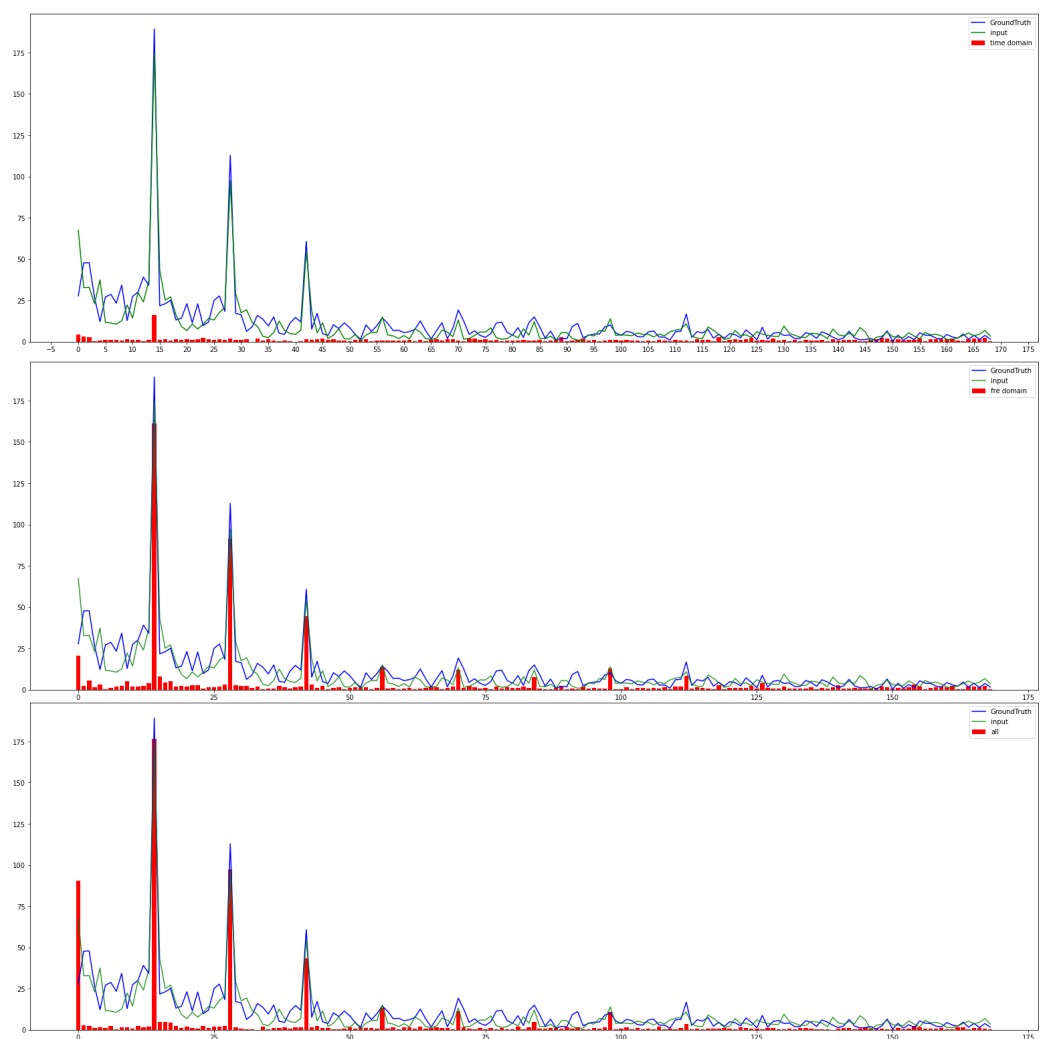

Figure 12: Discrete Fourier Transform (DFT) visualizations of input observations, predicted time series, and ground truth between recent Transformer-based approaches and our proposed method for Figure 11.

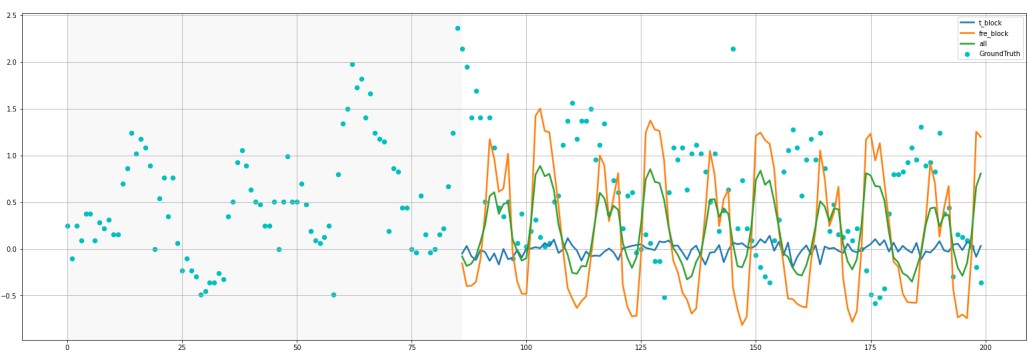

Figure 13: We visualized the outputs of T-Net, FreCoformer, and the final fused output separately.

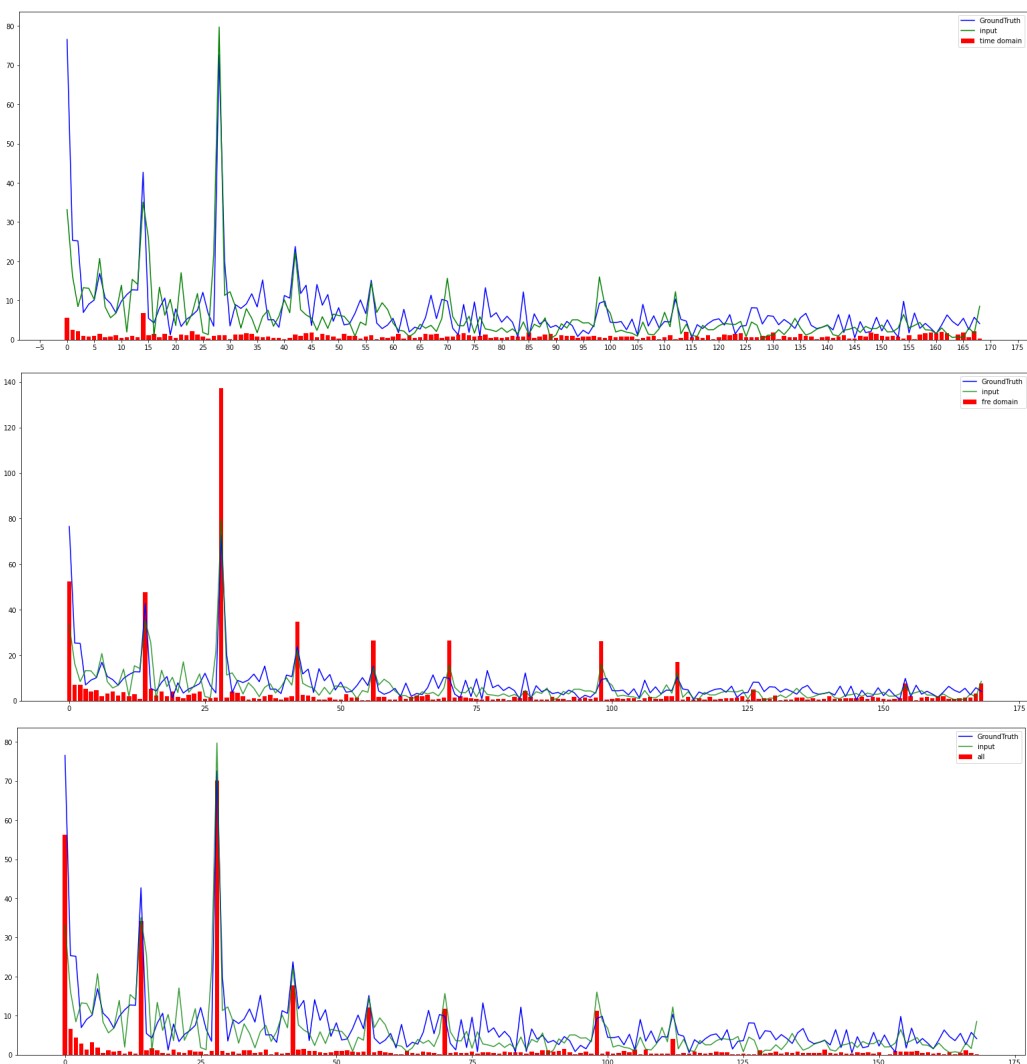

Figure 14: Discrete Fourier Transform (DFT) visualizations of input observations, predicted time series, and ground truth between recent Transformer-based approaches and our proposed method for Figure 12.

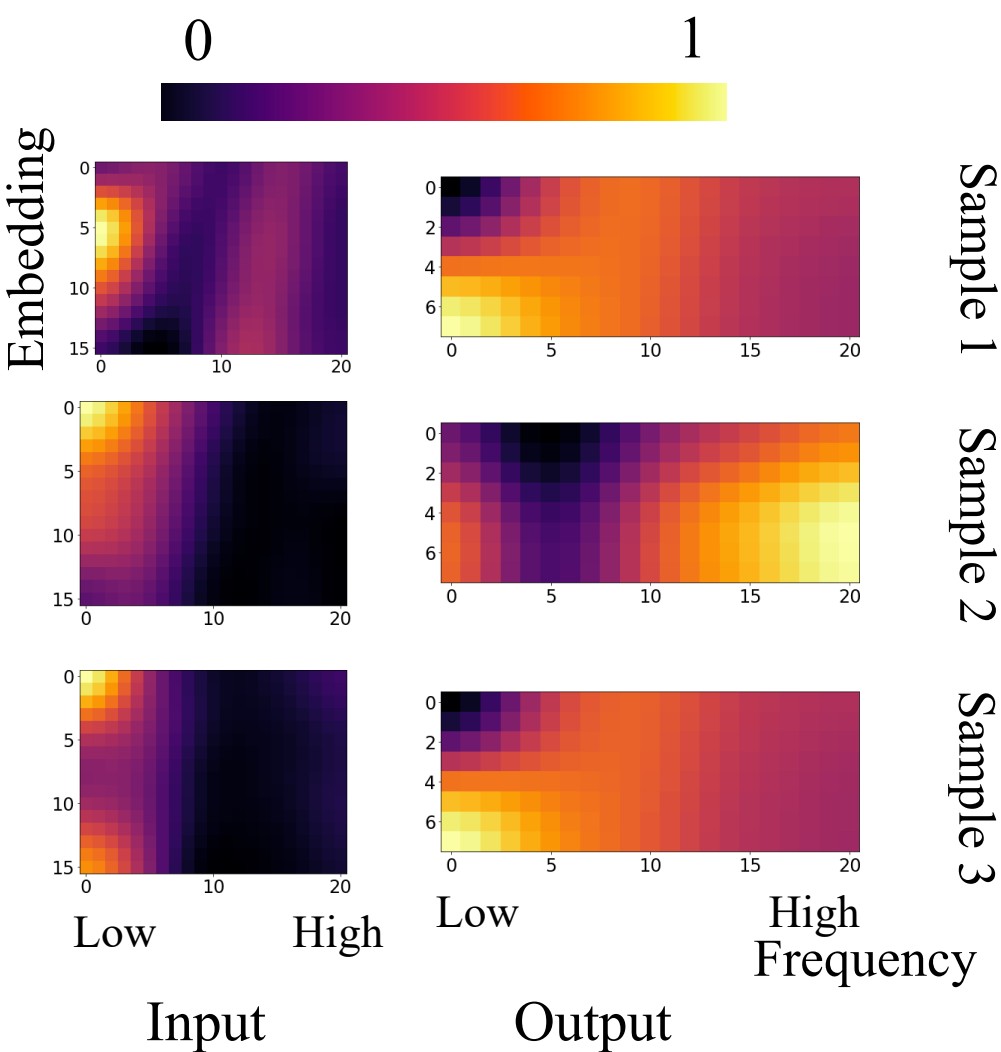

Figure 15: Heatmaps of the input and output matrixes of FreCoformer's Transformer encoder on Weather, we showed 3 samples from different channels. These output matrixes will be used to generate forecasting. The X-axis denotes frequency components, Y-axis is the dimension of the feature vector. These heat maps show the energy distribution in the frequency domain.

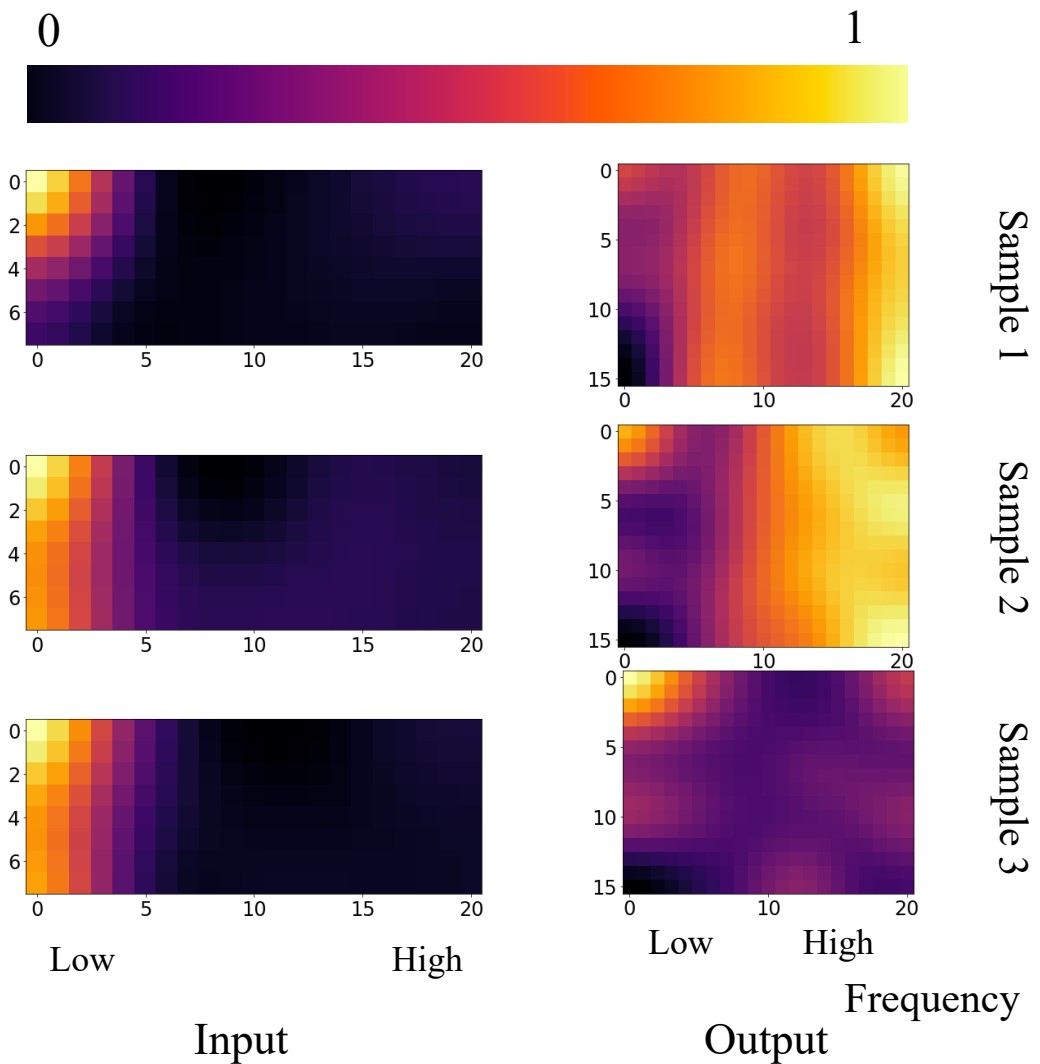

Figure 16: Heatmaps of the input and output matrixes of FreCoformer's Transformer encoder on ETTm1, we showed 3 samples from different channels. These output matrixes will be used to generate forecasting. The X-axis denotes frequency components, Y-axis is the dimension of the feature vector. These heat maps show the energy distribution in the frequency domain.

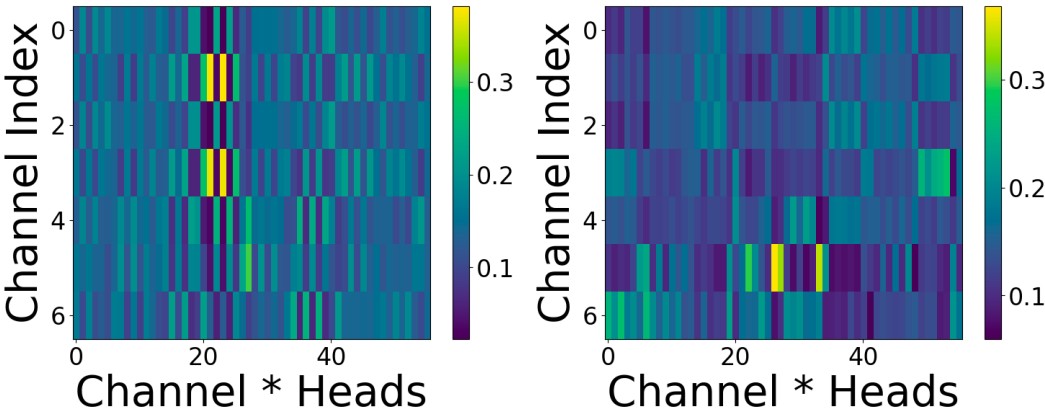

Figure 17: Heatmaps of the self-attention matrix of our proposed method. The left graph illustrates a low-frequency case, while the right graph represents a high-frequency case. The horizontal and vertical axes of the graph represent different channels. This figure demonstrates that our method can independently learn different channel-wise features for each frequency band.