# OpenReview forum: "Divide-and-Conquer Time Series Forecasting with Auto-Frequency-Correlation via Cross-Channel Attention"
_ICLR.cc/2024/Conference — Submitted to ICLR 2024_

### Official Review · Reviewer_xqFM · 2023-10-22

**Soundness:** 2 fair
**Presentation:** 2 fair
**Contribution:** 2 fair
**Rating:** 5
**Confidence:** 3

**Summary:**

This work presents a divide-and-conquer framework, FreCoformer, which is designed on top of the frequency domain for time series forecasting. Specifically, the framework comprises three key designs: frequency patching operation and two independent observations of these patches, and the authors further introduce a simple linear projection-based module to incorporate the information from the time domain. Extensive experiments show the effectiveness of the proposal method.

**Strengths:**

1. The evaluation is extensive. The authors validate the proposed method on multiple datasets and compare it with recent competing methods which prove the effectiveness of the proposed method. Moreover, the authors conduct adequate ablation to validate each design.
2. Results seem promising. The proposed method FreCoformer achieves strong performance over multiple datasets compared with existing works.
3. Writing is clear. Overall, the writing is clear and it is easy to understand each module presented by the authors.

**Weaknesses:**

1. Design of the framework. The divide-and-conquer framework seems to be a multiple path/view framework that has been extensively studied in the previous literature and it is not surprising that FreCoformer obtains competitive performance as it utilizes the information from both the time domain and frequency domain.
2. Training/inference efficiency. The following question from point 1 is how is the training/inference efficiency of FreCoformer compared to other competing methods as it computes on different domains. The authors have only compared the GPU memory between FreCoformer and other methods, which is not the most important metric to evaluate the efficiency. For example, training time/number of parameters/FLOPs/GPU latency are commonly used metrics for efficiency evaluation.
3. Core contribution. The authors have proposed lots of modules and designs in this work that may deviate from the core of 'divide-and-conquer'. For example, NYSTROM-FRECOFORMER just simply adopts the idea from NystromFormer and can hardly be claimed as the contribution of this paper. It is advised the authors rethink what in essence they are trying to present.

**Questions:**

1. Different channels for different datasets. From Tab. 2, It can be noticed that the authors use different channels for different datasets and the reviewer wonders the reasons behind this. Is it caused by different data lengths?

---

> ### Author Response · Authors · 2023-11-18
> **Reply to reviewer xqFM**
>
> We thank the reviewer for their efforts and time in reviewing our paper. Also, we appreciate the reviewer's perception of our contributions. We provided detailed responses to the reviewer's comments/questions as follows.
>
> Addressing the reviewer’s comments on weakness:
>
> (1) Thank you again for the valuable comments.
> In this paper, we would like to provide a "divided-and-conquer" modeling strategy for time series forecasting studies.
> This strategy is the spirit of the proposed framework.
> That is, we argue a successful forecasting approach might be designed on top of scenarios-specialized modules to capture eventful characteristics, rather than a unified framework.
> Please kindly refer to the Reviewer-cikd's (the first reviewer) reply.
>
> (2) Thank you for your question regarding our choice to use GPU memory usage as a metric in our study. Our decision is informed by related works in the field, especially in works focusing on transformer-based works [1].
> In recent years, a significant challenge has been their high demand for computational memory. The core context of the Transformer is the attention calculation that involves a number of matrix multiplications and matrix storages.
> This is essentially relevant when processing long input sequences. That is, low memory cost allows us to use longer time series observations for improving forecasting accuracy [4].
>
>
> (3) We would summarize our contributions here:
>
> 1) FeCoformer for enhancement of capturing frequency features;
>
> 2) Divide-and-conquer modeling strategy;
>
> 3) As we mentioned in (2), memory cost is one of the main focuses in time series forecasting.
> Existing methods address this challenge using architecturally complicated Transformer-based model designs [1, 2].
> This direction often decreases the forecasting performance.
> Namely, recent work [4] uses a very simple linear model that can outperform all of the previous models on a variety of common benchmarks, and it challenges the usefulness of Transformer for time series forecasting.
> Our work does not follow the line in this direction, and we draw attention to the computation mechanism of the Transformer, i.e., the matrix issues in memory cost. We hence introduce matrix approximation methods to address this challenge without any architectural care.
> This attempt opens a new possibility for time series forecasting.
>
>
> Addressing the reviewer’s comments on the question:
>
> The "\#channel" refers to the total number of channels in the dataset. These mainstream time series datasets consist of multiple channels, ranging in quantity from 7 to 862. We use all channels presented in each dataset without selection operation.
>
>
> [1]
> Zhou et al., Informer: Beyond Efficient Transformer for Long Sequence Time-Series Forecasting, 2021, AAAI
>
> [2]
> Zhou et al., FEDformer: Frequency Enhanced Decomposed Transformer for Long-term Series Forecasting, 2022, ICML
>
> [3]
> Nie et al., A Time Series is Worth 64 Words: Long-term Forecasting with Transformers, 2023, ICLR
>
> [4]
> Zeng et al., Are Transformers Effective for Time Series Forecasting? 2023, AAAI

---

> > ### Comment · Reviewer_xqFM · 2023-11-19
> > **Reply to authors**
> >
> > Thanks for the reply from the authors, but my concerns still remain. Especially:
> >
> > (1) the unclear core contribution: the proposed method seems to be a dual version of the baseline method PatchTST, and many proposed modules share the same designs with existing works which hurts the originality of this paper.
> >
> > (2) extra computational and parameter costs brought by dual-branch design: since FreCoformer adopts a multiple path/view framework, it is expected to bring more computational and parameter costs compared to the single branch version, which makes the improvement upon baseline methods less surprising.
> >
> > Therefore, the reviewer decides to keep this rating.

---

### Official Review · Reviewer_wrqE · 2023-10-25

**Soundness:** 3 good
**Presentation:** 3 good
**Contribution:** 2 fair
**Rating:** 3
**Confidence:** 3

**Summary:**

This paper proposes a new model, FreCoformer, for time series forecasting which operates on frequency and time domains. In detail, FreCoformer incorporates modules computing frequency and inter-feature correlation with patchifying frequencies. Furthermore, the divided-and-conquer method is introduced to tackle varying data scenarios. Finally, with Nystrom approximations, it achieves efficient linear complexity when considering inter-feature dependencies. As a result, FreCoformer achieves 41 top-1 and 21 top-2 cases out of 64 in total experimental settings for time series forecasting, showing the most efficient cost.

**Strengths:**

**Good inspiration about high-frequencies parts**
This paper gives visualization results that show that existing models operating on the time domain don't capture high-frequency dynamics. In contrast, models on the frequency domain, such as FEDformer and FreCoformer, are good at catching short-term variations. This gives new research direction to time series forecasting tasks.

**Weaknesses:**

**Insufficient novelty and contributions**
1. I think there are the following contributions in this paper: patching frequencies and cross-feature and cross-frequency self-attention in  Transformers operating on frequency domains, Divide-and-Conquer method, and Nyström-FreCoformer. However, cross-feature and cross-frequency self-attention are almost similar to [1], [2], and [3]. Also, the Divide-and-Conquer method is quite similar to a variant of PatchTST [4] where a transformer part for temporal connections is changed into a linear one. Also, Nyström-FreCoformer is almost the same as Nyströmformer [5]. As for the patching method on frequency domains, just domain is different from that of [4]. Therefore, I think that this paper just combines existing methods, and something new is not introduced. Therefore, to make more contributions, the authors have to propose stronger reasons why they combine such modules, or different parts from [1-5].

**Insufficient explanation or experimental results for argument**
1. According to 'Abstract', FreCoformer is designed for short-term variation while the divide-and-conquer framework is for long-term and short-term temporal variations. Can you give intuitive empirical evidence for this argument?

2. In channel-wise attention, you design self-attention differently from the original one [6]. Why do you select your own design? Can you give some reason for this selection?

3. In Section 3.3, the authors say that "To compute the attention matrix A, we first select m landmark columns from the input Qn and Kn matrices in each channel.". Can you give the detailed procedure to select $m$ landmark columns from entire ones?

4. In Figure 3 (a), the authors say that "Compared to the best-performing PatchTST, our model exhibits an advantage in identifying short-term variations, resulting in detailed fluctuations in periodicity variation." However, I don't know what parts show these results. Can you mark what parts provide this information?

5. Additionally, there are some confusing parts. Refer to Question Section.

[1] Zhou et al., FEDformer: Frequency Enhanced Decomposed Transformer for Long-term Series Forecasting, 2022, ICML
[2] Chen et al., A Joint Time-frequency Domain Transformer for Multivariate Time Series Forecasting, 2023
[3] Zhang et al., Crossformer: Transformer Utilizing Cross-Dimension Dependency for Multivariate Time Series Forecasting, 2023, ICLR
[4] Nie et al., A Time Series is Worth 64 Words: Long-term Forecasting with Transformers, 2023, ICLR
[5] Xiong et al., Nyströmformer: A Nyström-Based Algorithm for Approximating Self-Attention, 2021, AAAI
[6] Vaswani et al., Attention is All You Need, 2017, Neurips

**Questions:**

1. In 'Abstract', the authors argue that "The patching process refines the frequency information, enhancing the locality.". What kind of locality is enhanced by the patching process? Locality on the time domain is quite intuitive but one on the frequency domain is not. I think that you have to elaborate on locality on the frequency domain and why we have to utilize the frequency locality.

2. In 'Abstract', there is a sentence saying "The subsequent observations extract the consistent representation within different channels by attention computation and summarize the relevant sub-frequencies to identify eventful frequency correlations for short-term variations.". What does 'Consistent representation within different channels' mean? I think that attention is capable of discovering relationships between different channels, not consistent representations within them.

3. In 'Introduction', there is a sentence saying "This approach relies on heuristic and empirical strategies, i.e., random or top-K
frequency selection, often capturing spurious correlations for forecasting (seen in Figure 1(c)).". I think this is a too strong argument because Figure 1 doesn't directly provide evidence for spurious correlations.

4. "These independent attentions share model parameters across all sub-frequency learning, preventing winner-take-all of redundancy low-frequency components.": About this sentence, Can you explain further? By just sharing the same Transformer's parameters across all sub-frequency learning, why is the phenomenon prevented that low-frequency components dominate against high-frequency ones?

5. "To demonstrate the efficacy of the core design—channel-wise attention in FreCoformer, we visualized the heatmaps of the input and output DFT matrices of the Transformer encoder in FreCoformer in Figure 3(b)." in Section 4.2.2: As far as I know [1], the DFT matrix is like a weight matrix to transform time domain into frequency domain. If the input length $N$ is the same, the DFT matrix always has the same value. What does 'DFT matrices' in the above sentence mean?

6. "This balance likely enables our method to efficiently extract pivotal frequency features across the entire frequency spectrum and various temporal variations, enhancing prediction outcomes." in Section 4.2.2: Why is balanced energy across entire frequencies helpful for your method? Can you give me more direct evidence for this sentence?


[1] https://en.wikipedia.org/wiki/DFT_matrix

---

> ### Author Response · Authors · 2023-11-18
> **Reply to reviewer wrqE [part 1]**
>
> We appreciate your time and detailed review, and we also thank you for the reviewer's recognition of the strength of this paper.
> To address the reviewer's concerns/comments, we provide the following detailed responses.
>
> First, we would like to address the overall goal and contribution of the paper:
>
> From a data perspective, we observed that real-world time series exhibit variability across different scenarios.
> Such variations can be divided into short-term and long-term variations in time series forecasting.
> We argue that a successful forecasting approach might be designed on top of scenarios-specialized modules to capture eventful characteristics, rather than a unified model that solely operates at either the time domain (like Informer [1] and PatchTST [2], Crossformer [5]) or the frequency domain (FEDformer [3] and ETSformer [4]).
> We hence propose this divided-and-conquer modeling strategy.
>
> From a model perspective, we would enhance the capacity to capture short-term temporal variation.
> We hence draw our attention to the frequency domain, since models operated on the time domain practically extract the characteristics of a combination of various frequency components.
> This paper proposes FreCoformer to reveal what frequencies are present in a time series scenario, and in what proportions.
>
> Further, this investigates a new solution for tackling the common memory cost challenge.
> Different from existing solutions that meticulously design a lightweight model architecture, we draw our attention to the computation mechanism of the Transformer, i.e., the matrix multiplication and storage arising memory cost.
> We hence introduce matrix approximation methods to address this challenge without any architectural care [6].
>
> Addressing the reviewer’s comments on the weakness:
>
> (1) As shown in Table 4(Left) of the main body, the two modules contribute differently to the final prediction on distinctive datasets. On the Weather dataset, T-Net is the main contributor (all perform second-best results), and on the ETTh1 dataset, FreCoformer becomes more important.
> A similar conclusion can be found in Figures 1, and 2 of the newly uploaded PDF.
> These results show that our framework can automatically capture different features among different datasets.
> Combining both modules mainly indicates better performance.
>
> (2) I think we keep the consistency of the original one and ours. We just deploy this self-attention on two different settings.
>
> (3) In our implementation of the Nystromformer model [6], the selection of $m$ landmarks is determined through the process described in the following:
>
> The process involves dividing the input sequence into equal-sized segments and then summarizing each segment to form a landmark. Specifically, for a sequence of length $n$, we calculate the stride length $l$ as the ceiling of $n/m$ and then perform a sum reduction over each segment of length $l$ in the query and key matrices.
> This results in $m$ landmarks that represent the original sequence.
>
> (4) This result can be found in Fig.3(a) of the main body.
>
> Among these comparisons, PatchTST (blue line) and ours (red line) have satisfied data fit (dodgerblue discrete points).
> Compared to PatchTST, our method can further capture some local fluctuations upon the periodic changes.
>
> [1]
> Zhou et al., Informer: Beyond Efficient Transformer for Long Sequence Time-Series Forecasting, 2021, AAAI
>
> [2]
> Nie et al., A Time Series is Worth 64 Words: Long-term Forecasting with Transformers, 2023, ICLR
>
> [3]
> Zhou et al., FEDformer: Frequency Enhanced Decomposed Transformer for Long-term Series Forecasting, 2022, ICML
>
> [4]
> Woo et al., ETSformer: Exponential Smoothing Transformers for Time-series Forecasting, 2022, arXiv
>
> [5]
> Zhang et al., Crossformer: Transformer Utilizing Cross-Dimension Dependency for Multivariate Time Series Forecasting, 2023, ICLR
>
> [6]
> Xiong et al., Nyströmformer: A Nyström-Based Algorithm for Approximating Self-Attention, 2021, AAAI
>
> [7]
> Cuaresma et al., Forecasting electricity spot-prices using linear univariate time-series models, 2004, Applied Energy
>
> [8]
> Thompson et al., Multifractal detrended ﬂuctuation analysis: Practical applications to ﬁnancial time series, 2016, Mathematics and Computers in Simulation
>
> [9]
> Hammond et al., High-frequency sensor data capture short-term variability in Fe and Mn concentrations due to hypolimnetic oxygenation and seasonal dynamics in a drinking water reservoir, 2023, Water Research
>
> [10]
> Chen et al., A Joint Time-frequency Domain Transformer for Multivariate Time Series Forecasting, 2023

---

> ### Author Response · Authors · 2023-11-18
> **Reply to reviewer wrqE [part 2]**
>
> Addressing the reviewer’s comments on the questions:
>
> Question 1:
>
> As we know, low-frequency oscillation represents the long-term variation or periodicity, and the frequency locality usually refers to the short-term variations, that is, high-frequency components.
> As we described in section 3.2, real-world time series exhibit variability across different scenarios.
> Sometimes, areas like banking transactions, electricity consumption, and hospital foot traffic [7, 8, 9] require a focus on short-term variations.
> In our ablation study, Table 4(Right, "Non-FP"), and in Figure 5 of the uploaded PDF, we compare the outcomes of processing the entire frequency domain as a whole versus segmenting it into independent patches.
> The results, both in terms of predictive accuracy and the visualization of frequency information, clearly show the effectiveness of locality enhancement.
>
> Question 2:
>
> In the multi-channel time series, some of the channels may exhibit consistent correlations within specific frequency bands(seen in Fig.4 in the new updated PDF).
> We would use channel-wise attention to learn and understand these correlations and key frequency components, which appear in both observation and forecasting.
>
> Question 3:
>
> Except for ignoring some high frequencies, in Fig.1, there are some spurious components, where many components (red bar) weigh significantly higher than the ground truth and input data.
> A similar result can be found in Figure 4 of our uploaded PDF.
> We consider the representation performance of such random or Top-K selection methods (FedFormer [3], ETSformer [4], and JTFT [10]) more dependent on the selected offers to the model.
> This cannot treat the frequency components equally and often needs to be more balanced with some selection.
>
> Question 4:
>
> Through frequency band independence, our model (i)independently normalize inputs across different frequency bands and (ii)learns features within each band by learning the consistency of different channels.
> This approach ensures that the model allocates equal attention to all frequency bands, rather than predominantly focusing on the low-frequency components.
>
> Question 5:
>
> It is true that the left column is the original DFT transformation matrix.
> But the right column shows the resulting DFT transformation matrics by Transformer, that is, the output of the Transformer.
>
> Question 6:
>
> We show some experiments with/without frequency patching in Figure 5 of the uploaded PDF.
> We empirically consider the patch, and independent learning does not draw any attention to any other frequency component.
> This prevents winner-take-all of redundancy low-frequency components.
> The purpose of this paper is to provide such a strategy and to begin the investigation into its properties of frequency enhancement, knowing that it will take several papers to properly investigate this line of work.
>
> (The references can be found in part 1)

---

> > ### Comment · Reviewer_wrqE · 2023-11-20
> > **Thank you for your effort to resolve my concerns**
> >
> > Thank you for your effort to resolve my concerns.
> >
> > However, I think that a large elaboration has to be done on this paper. My major concerns consist of the reason for the authors' arguments. I think that when proving some arguments with not some strong intuition or theories but empirical observations, many experiments have to be done for justification. However, the reasons the authors propose are quite insufficient. Also, about my concerns, major changes are not made in the manuscript. Therefore, I think this paper requires significant changes and I uphold my scores as before.

---

### Official Review · Reviewer_BtQH · 2023-10-31

**Soundness:** 3 good
**Presentation:** 2 fair
**Contribution:** 3 good
**Rating:** 6
**Confidence:** 4

**Summary:**

This work proposes a transformer-based architecture called FreCoformer for timeseries forecasting. FreCoformer is designed with patching operation in the frequency space, which gives good locality through maintaining local frequency information. The authors also propose a two-stage linear projection to incorporate time information into the network. To reduce the computation cost of the framework, the authors further used the matrix approximation in NystromFormer in the proposed network. The authors performed extensive experiments to demonstrate the effectiveness of the proposed architecture.

**Strengths:**

- This is solid work validating that frequency-space operations can be used to achieve good performance in the timeseries forecasting task.
- The authors make a lot of attempts to improve the performance of the proposed architecture while reducing the computation costs.
- Extensive experimental results with solid baselines and fair comparison.

**Weaknesses:**

- Why the “divide and conquer” framework is named “divide and conquer”? Does it have any correlation with the “divide and conquer” algorithm? Isn’t it just a two-layer linear operation? How similar it is to the linear aggregation operation as proposed in [1]?
- Since the frequency-domain signal essentially is a one-to-one projection of the time-domain signal, how much does the proposed network differ from PatchTST (theoretically)? Specifically, do the two architectures share the same, or a subset of solution space? The performance gain over PatchTST seems very marginal in Table 3.

A lot of typos, e.g. Attentione

[1] Liu, Ran, Mehdi Azabou, Max Dabagia, Jingyun Xiao, and Eva Dyer. "Seeing the forest and the tree: Building representations of both individual and collective dynamics with transformers." Advances in neural information processing systems 35 (2022): 2377-2391.

**Questions:**

NA

---

> ### Author Response · Authors · 2023-11-18
> **Reply to reviewer BtQH**
>
> We appreciate the reviewer's recognition of our contribution.
> We respond to the comments/concerns from the reviewer as follows.
>
> Weakness 1:
> Thank you again for the valuable comments.
> In this paper, we would like to provide a "divided-and-conquer" modeling strategy for time series forecasting studies.
> It consists of the FreCoformer and the design of a linear module.
> Real-world time series exhibit variability across different scenarios.
> Therefore, using two modules, our "divide and conquer" framework is designed to capture long-term and short-term variations respectively.
> This strategy is the spirit of the proposed framework.
> That is, we argue a successful forecasting approach might be designed on top of scenarios-specialized modules to capture eventful characteristics, rather than a unified framework.
> Please refer to the Reviewer-CiKd's (the first reviewer) reply for more experimental evidence.
>
> The paper [1] is architecturally somehow related to our design.
> However, all the operations are deployed on the time domain.
> This difference is the main focus of this paper, that is,
> models operated on the time domain extract the characteristics of a combination of various frequency components.
> Time series forecasting would know what frequencies are present in our signal and in what proportions.
> Frequency analyses can highlight eventful characteristics, such as key neural waveforms or electrical activities in paper [1].
>
>
>
>
> Weakness 2:
> As mentioned below, although DFT projects an observation of the time domain to the frequency domain, a Fourier transform of a signal tells you the frequency composition.
> PatchTST [2] uses patching on the time domain and aims to learn time-time correlations for forecasting. However, it results in information loss of high frequency, as shown in Figure 1 of the main body.
>
> On the other hand, our method focuses on capturing the important frequency feature, by learning features within each band independently. Through this, FreCoformer can capture more apparent middle-to-high-frequency features related to short-term variations in time series data.
> Our forecasting results outperform PatchTST, particularly on datasets with a rich spectrum of frequency features, such as ETTh1. The improvement in our predictive performance is significant.
>
>
> [1] Liu et al., Seeing the forest and the tree: Building representations of both individual and collective dynamics with transformers., 2022, Advances in neural information processing systems 35.
>
> [2]
> Nie et al., A Time Series is Worth 64 Words: Long-term Forecasting with Transformers, 2023, ICLR

---

### Official Review · Reviewer_CiKd · 2023-11-01

**Soundness:** 2 fair
**Presentation:** 2 fair
**Contribution:** 2 fair
**Rating:** 6
**Confidence:** 3

**Summary:**

**Summary:**

This paper introduces the FreCoformer, a novel Transformer-based model tailored for capturing short-term temporal variations in time series data. The model is distinguished by its frequency patching operation, attention mechanisms for extracting consistent representations across different channels, and a divide-and-conquer framework for learning both long-term and short-term temporal variations. The paper also proposes a lightweight variant of the FreCoformer, employing the Nystrom method to reduce parameters and computational costs.

**Strengths:**

1. Innovative Approach: The application of patching in the Fourier domain combined with the use of Nystromformer is a novel contribution to the forecasting domain.

2. Empirical Performance: The proposed model outperforms PatchTST in most experimental settings, indicating its effectiveness.

**Weaknesses:**

1. From my perspective, the term *divided-and-conquer* is a little bit over claim. I would expect the divided and conquer type model to have a tree structure or multi-scale design. The authors couple an attention block within the Fourier domain and an MLP-type block for the first-order difference sequence, which, in my mind, is just kind of a dual structure. If the authors think divided-and-conquer term indeed reflects the spirit of the proposed model, it would be better to add more discussions on it.

2. The test data loader in the provided sample codes sets `drop_last = True`, leading to the exclusion of several test samples. Correcting this and rerunning the experiments would enhance the validity of the results.

3. The random control experiments seem not included. The model variance and hyperparameter-sensitive analysis may help to further highlight the efficiency of the proposed model.

**Questions:**

1.  The paper describes using Attention in the Fourier domain and MLP in the time domain. Could the authors provide more insight into this specific design choice? Why not use Attention layers or MLPs in both domains?

2. In the code, the time domain considers both trend differences and local differences. However, in section 3.2 of the main paper, only the first-order differences are mentioned. Does it only correspond to the local differences or both trend/local differences?

At the current stage, I tend to recommend accepting this paper if the numerical experiment results are updated. However, my final decision is open to change pending the authors' rebuttal and further discussions with other reviewers and the Area Chair.

**Strengths:**

Please see the Strengths section in Summary.

**Weaknesses:**

Please see the Weaknesses section in Summary.

**Questions:**

Please see the Questions section in Summary.

---

> ### Author Response · Authors · 2023-11-18
> **Reply to reviewer CiKd [part 1]**
>
> Thank you again for the valuable comments. Indeed, we would like to provide a modeling strategy, "divided-and-conquer," for time series forecasting studies.
> We argue a successful forecasting approach might be designed on top of scenarios-specialized modules to capture eventful characteristics.
>
> From a data observation, real-world time series exhibit variability across different scenarios. As we introduced in section 3.2, the main body of the paper, such variations can be divided into short-term and long-term variations. We include temporal visualization on two cases of datasets in Figure 1 of the uploaded PDF. It shows apparent differences in temporal variations between these two cases.
> Existing methods usually develop a unified model to capture these variations.
> Also, they solely operate at either the time domain (e.g., Informer [1] and PatchTST [2]) or the frequency domain (FEDformer [3] and ETSformer [4]).
> We observed such methods often lead to information loss, especially short-term variations, that is, high-frequency components shown in Figure 1 of the main body.
>
> Therefore, using two modules, our "divide and conquer" framework is designed to capture long-term and short-term variations independently.
> Meanwhile, they are complementary to each other.
> As shown in Table 4(Left) of the main body, the two modules contribute differently to the final prediction on distinctive datasets. On the Weather dataset, T-Net is the main contributor (all perform second-best results), and on the ETTh1 dataset, FreCoformer becomes more important.
> A similar conclusion can be found in Figure 2 of the uploaded PDF.
> These results show that our framework can automatically capture different features among different datasets.
> Combining both modules mainly indicates better performance.
>
> Random control experiments:
>
> Thank you for your valuable suggestion.
> In response, we have added the experiment results in Figure 3 of the uploaded PDF.
> This figure illustrates the results of experimenting with 200 different combinations of parameters on two datasets: the Weather dataset, characterized by predominantly low-frequency features, and the ETTm2 dataset, notable for its high-frequency features.
> Practically, this paper focuses on a parameter combination that consistently demonstrates stability, rather than selecting the combination that achieves the highest accuracy.
>
> For the "drop\_last = True" issue and random control experiments, please check part 2.
>
> Q1: Thank you for the interesting question.
>
> We would develop two effective modules that could extract different features while keeping their structures as simple as possible.
>
> In the temporal domain, MLPs are effective due to their ability to summarize long-term temporal dependencies.
> The full connection computation allows MLPs to consider all input features simultaneously, extracting global semantic information of input data.
> It is shown in the recent paper [5] that a very simple MLP model can outperform all of the previous models on a variety of common benchmarks.
>
> In contrast, the frequency domain presents a set of challenges. We consider Transformer could enhance the locality (patching) and learn the correlation of these local features (attention). In our design, the frequency domain module benefits from this ability, by identifying important frequency features across channels.
>
> Q2: Yes, it corresponds to both. The two functions in the code both refer to local differences.
> In the paper, 'global' and 'local' refer to the 2-layer MLP architecture. In our code, we used 2 inputs for local information extraction: "local" differences and "trend" differences. Both are aimed at capturing local dynamics within the time domain.
> Also, the first-order differences mentioned in section 3.2 primarily refer to these local aspects, encompassing variations both within and between patches.
>
> [1]
> Zhou et al., Informer: Beyond Efficient Transformer for Long Sequence Time-Series Forecasting, 2021, AAAI
>
> [2]
> Nie et al., A Time Series is Worth 64 Words: Long-term Forecasting with Transformers, 2023, ICLR
>
> [3]
> Zhou et al., FEDformer: Frequency Enhanced Decomposed Transformer for Long-term Series Forecasting, 2022, ICML
>
> [4]
> Woo et al., ETSformer: Exponential Smoothing Transformers for Time-series Forecasting, 2022, arXiv
>
> [5]
> Zeng et al., Are Transformers Effective for Time Series Forecasting? 2023, AAAI

---

> ### Author Response · Authors · 2023-11-18
> **Reply to reviewer CiKd [part 2]**
>
> For the "drop\_last = True" issue:
>
> We thank the reviewer for the detailed review. This setting is for keeping the batch size consistent.
> When set to True, it instructs the data loader to drop the last incomplete batch if the dataset size is not divisible by the batch size. We here show the comparison of “True” and “False” settings in the Table below.
> As we can see, there is almost no difference.
>
> | Metrics    | MSE (drop last = True) | MAE (drop last = True) | MSE (drop last = False) | MAE (drop last = False) |
> |------------|------------------------|------------------------|-------------------------|-------------------------|
> | ETTh1_96   | 0.362                  | 0.391                  | 0.363                   | 0.392                   |
> | ETTh1_192  | 0.403                  | 0.411                  | 0.403                   | 0.411                   |
> | ETTh1_336  | 0.406                  | 0.415                  | 0.405                   | 0.415                   |
> | ETTh1_720  | 0.433                  | 0.452                  | 0.432                   | 0.450                   |
> | Weather_96 | 0.149                  | 0.196                  | 0.147                   | 0.196                   |
> | Weather_192| 0.193                  | 0.238                  | 0.192                   | 0.239                   |
> | Weather_336| 0.245                  | 0.279                  | 0.246                   | 0.280                   |
> | Weather_720| 0.318                  | 0.332                  | 0.319                   | 0.332                   |
>
> Random seeds:
>
> For random seed experiments, we selected two data sets, ETTh1 and Weather. The selection of random numbers is 2019, 2020, 2021, 2022, and 2023. The table records the mean value and fluctuation range of the results under various experimental settings with different seed selections:
>
> | Metric |     | MSE             | MAE             |
> |--------|-----|-----------------|-----------------|
> | Weather| 96  | 0.1477 ± 0.0008 | 0.1959 ± 0.0011 |
> |        | 192 | 0.1921 ± 0.0003 | 0.2394 ± 0.0010 |
> |        | 336 | 0.2448 ± 0.0009 | 0.2803 ± 0.0009 |
> |        | 720 | 0.3209 ± 0.0007 | 0.3329 ± 0.0009 |
> | ETTh2  | 96  | 0.2743 ± 0.0004 | 0.3369 ± 0.0008 |
> |        | 192 | 0.3366 ± 0.0010 | 0.3774 ± 0.0009 |
> |        | 336 | 0.3242 ± 0.0008 | 0.3798 ± 0.0003 |
> |        | 720 | 0.3752 ± 0.0006 | 0.4208 ± 0.0010 |
> | ETTm1  | 96  | 0.2844 ± 0.0005 | 0.3381 ± 0.0004 |
> |        | 192 | 0.3226 ± 0.0007 | 0.3624 ± 0.0005 |
> |        | 336 | 0.3539 ± 0.0002 | 0.3840 ± 0.0002 |
> |        | 720 | 0.4097 ± 0.0009 | 0.4208 ± 0.0005 |

---

### Author Response · Authors · 2023-11-18
**Summary of Revisions**

First and foremost, we would like to express our gratitude to all the reviewers for their insightful and valuable comments.
We also appreciate the positive feedback and recognition of our paper.
After carefully considering the reviewers' suggestions, we have updated our supplementary materials with some new charts to further facilitate the understanding of our paper.
This supplementary material will be referenced in our responses to the reviewers' comments.

These new charts, spanning 3 pages, have been added before the original appendix section. They primarily include:

* Figure 1: A visualization comparison of the prediction results of our time-domain and frequency-domain modules across 2 different datasets.

* Figure 2: A frequency-domain comparison of the prediction results of our time-domain and frequency-domain modules on 2 different datasets.

* Figure 3: The accuracy results of our framework across 200 different random sets of hyperparameters, along with a case study for some hyperparameters.

* Figure 4: A visualization in the frequency domain of the prediction results of FEDformer [1] and our method on multiple channels of the ETTh1 dataset.

* Figure 5: A visualization in the frequency domain of the prediction results obtained with and without the use of frequency domain patching.

In addition, some typos in the manuscripts have also been corrected.

The reviewers' feedback is immensely valuable to us, and we sincerely look forward to any further questions.

[1]
Zhou et al., FEDformer: Frequency Enhanced Decomposed Transformer for Long-term Series Forecasting, 2022, ICML

---

### Meta-Review · Area_Chair_ETKZ · 2023-12-06

**Metareview:**

The paper proposes a time series forecasting model based on Transformers, with modules that learn short-term and long-term temporal variations by leveraging both the time and frequency domains. The reviewers appreciated the idea of using Nystromformer + frequency modeling, and the extensive experiments, as well as the consideration given to computational cost. However, the issue of insufficient novelty was raised by two reviewers and Reviewer wrqE indicated a list of past work. The authors did not sufficiently explain the differences between their work and the existing methods. Moreover, the use of the term "divide and conquer" isn't justified in this context, it seems like a misnomer. The authors did address some of the concerns, such as the potential data issue raised by Reviewer CiKd, the random control experiments and the comparison to PatchTST. However, the insufficient differentiation compared to prior work and the less-than-clear title remain unsolved, and place the paper below the bar for acceptance.

**Justification For Why Not Higher Score:**

Authors have failed to convince the reviewers that the paper has significant contributions.

**Justification For Why Not Lower Score:**

N/A

---

### Decision · Program_Chairs · 2024-01-16

Reject